# TRANSFACE: UNIT-BASED AUDIO-VISUAL SPEECH SYNTHESIZER FOR TALKING HEAD TRANSLATION

## ABSTRACT

Direct speech-to-speech translation achieves high-quality results through the introduction of discrete units obtained from self-supervised learning. This approach circumvents delays and cascading errors associated with model cascading. However, talking head translation, converting audio-visual speech (i.e., talking head video) from one language into another, still confronts several challenges compared to audio speech: (1) Existing methods invariably rely on cascading, synthesizing via both audio and text, resulting in delays and cascading errors. (2) Talking head translation has a limited set of reference frames. If the generated translation exceeds the length of the original speech, the video sequence needs to be supplemented by repeating frames, leading to jarring video transitions. In this work, we propose a model for talking head translation, **TransFace**, which can directly translate audio-visual speech into audio-visual speech in other languages. It consists of a speech-to-unit translation model to convert audio speech into discrete units and a unit-based audio-visual speech synthesizer, Unit2Lip, to re-synthesize synchronized audio-visual speech from discrete units in parallel. Furthermore, we introduce a Bounded Duration Predictor, ensuring isometric talking head translation and preventing duplicate reference frames. Experiments demonstrate that our proposed Unit2Lip model significantly improves synchronization (1.601 and 0.982 on LSE-C for the original and generated audio speech, respectively) and boosts inference speed by a factor of ×4.35 on LRS2. Additionally, TransFace achieves impressive BLEU scores of 61.93 and 47.55 for Es-En and Fr-En on LRS3-T and 100% isochronous translations. [1]

## 1 INTRODUCTION

With the rapid advancement of online communication technology, the demand for talking head translation technology (KR et al., 2019; NVIDIA, 2022; Waibel et al., 2022; Song et al., 2022; Waibel et al., 2023) has surged across various domains including online meetings, education, and healthcare. The aim is to create cross-language talking heads that both the audio speech and the visual speech correspond to the content of the target language and keep the audio-visual synchronization. Presently, most works (Lavie et al., 1997; KR et al., 2019; Zhang et al., 2021; Waibel et al., 2023) is carried out in a cascading manner, entailing automatic speech recognition (ASR) (Yu & Deng, 2016; Potamianos et al., 2004) , neural machine translation (NMT) (Brown et al., 1990; Bahdanau et al., 2014), text-to-speech (TTS) (Klatt, 1987; Ren et al., 2019; 2020), and wav2lip (Prajwal et al., 2020) models working in succession. However, this approach accumulates significant errors and results in excruciatingly slow inference. Moreover, it falls short when dealing with text-less languages like Minnan and various dialects. Some speech-to-speech translation efforts (Wahlster, 2013; Lee et al., 2021; Huang et al., 2022; Nguyen et al., 2023) have introduced textless NLP, incorporating discrete units acquired through self-supervised learning (Schneider et al., 2019; Hsu et al., 2021) to represent the speech of the target language. It replaces text as the training target and enables direct speech-to-speech translation. (Lee et al., 2021).

Nevertheless, despite these advancements, the talking head translation models (KR et al., 2019; Song et al., 2022; Waibel et al., 2023) still rely on both text and audio speech for wav2lip processing. This incurs needless inference speed reduction and introduces additional cascading errors.

---

[1]Samples are available at `https://transface-demo.github.io/`

The current talking head translation framework primarily grapples with the following hurdles: **(1) Sluggish Inference and Cascading Error:** Existing methods, being built on the cascade model approach, necessitate the synthesis of talking heads through both text and speech. This results in a sluggish inference and an failure to provide additional supplementary visual information (e.g., visually distinguishable phonemes /ae/ and /ai/). **(2) Lack of Parallel Corpus Data:** The acquisition of visual corpus data is obviously more challenging compared to audio corpus, which requires full frontal videos. Parallel visual translation corpus pairs dataset for direct talking head translation is difficult to construct. **(3) Fixed Number of Reference Frames:** The translation outcomes can often be excessively lengthy, requiring the reuse of reference frames, which, in turn, leads to jarring video transitions.

To address these challenges, we propose a direct talking head translation framework, Transface, consisting of a speech-to-unit translation model (S2UT) to convert audio speech into discrete units and a unit-based audio-visual speech synthesizer (Unit2Lip) to re-synthesize synchronized audio-visual speech from discrete units in parallel. Unit2Lip achieves acceleration with parallel synthesis of audio and visual speech (Unit-To-Audio & Visual) instead of the traditional serial synthesis (Unit-To-Audio-To-Visual). Meanwhile, since the S2UT model can learn cross-linguistic mapping from parallel audio data, we directly apply this cross-linguistic mapping to the talking head translation to realize zero-shot Talking Head Translation without parallel visual corpus. Finally, to address the challenge of the fixed number of reference frames, we propose a Bounded Duration Predictor module, which can uniformly coordinate the number of frames sustained by each discrete unit according to the target length, and thus control the total length, realizing Isometric Talking Head Translation and improving the acceptance of the translation result.

The code and model will released and the main contributions of this paper are as follows:

- We introduce the first direct talking head translation framework, TransFace, capable of synthesizing talking heads without relying on audio speech and text. This innovation effectively circumvents the slowdown and cumulative error typically associated with model cascading.

- We propose the first unit-based audio-visual speech synthesizer, Unit2Lip, which can synthesize audio and visual speeches in parallel while maintaining synchronization with audio speech, achieving $4.35\times$ inference speedup.

- We propose a bounded-duration-predictor that achieves 100% isometric talking head translation, which is significant for streaming translation scenarios. Also, it effectively avoids jarring video transitions and improves the acceptance of translated talking head videos.

- We conduct experiments to demonstrate that Unit2Lip achieves a notable improvement in synchronization (1.601 LSE-C improvement for original speech and 0.982 LSE-C improvement for generated speech) on LRS2. Additionally, TransFace achieves 61.93 and 47.55 BLEU scores for Es-En and Fr-En on LRS3-T, respectively.

## 2 RELATED WORKS

### 2.1 AUDIO-VISUAL SPEECH SYNTHESIS.

The synthesis of high-quality audio and visual speech is an ongoing area of research, given its significance as an information carrier. In the early stages of development, researchers initially rely on mel spectrograms for resynthesizing audio-visual speech. WaveNet (Oord et al., 2016) first demonstrates that convolutional neural networks can synthesize high-fidelity audio speech from mel-spectrograms. With complex architectural considerations and adversarial generative networks, MelGAN (Kumar et al., 2019) and Hifi-GAN (Kong et al., 2020) achieves acceleration while synthesizing high-quality speech. Additionally, some studies (KR et al., 2019; Prajwal et al., 2020) are employing GAN networks to generate the corresponding audio-lip synchronized visual speech (talking head) from mel-spectrograms. With the advent of self-supervised learning (Schneider et al., 2019; Baevski et al., 2020; 2019), researchers are beginning to explore the resynthesis of audio speech from discrete units (Polyak et al., 2021; Lee et al., 2021). They are successfully achieving audio speech quality comparable to that of resynthesis from mel-spectrograms, demonstrating that discrete units contain sufficient information for effective audio resynthesis. Subsequent efforts (Polyak et al., 2021; Lee et al., 2021; Huang et al., 2022; Zhang et al., 2023; Seamless Communication, 2023) focus on a range of audio synthesis tasks rooted in discrete units. These models are

trained using autoregressive techniques, subsequently allowing for the resynthesis of corresponding audio speech from these discrete units. Discrete units reduce the complexity of modeling sequences and enhance the model's comprehension of sequences.

However, although audio-visual speeches are two temporally parallel media streams containing temporally consistent semantic information, no researchers have yet attempted to synthesis the talking head from discrete units. Currently, the existing talking heads can only be re-synthesized through audio speech, which significantly impacts inference speed and leads to the generated talking head lacking additional supplementary visual information. In this paper, we introduce a unit-based audio-visual speech synthesizer, which enables the direct talking head resynthesis from corresponding discrete units while maintaining perfect synchronization. The transition from the original sequential synthesis method (Unit-To-Audio-To-Visual) to a parallel synthesis method (Unit-To-Audio&Visual) leads to a substantial acceleration in the speed of audio-visual speech synthesis. Moreover, Unit2Lip does not have a dependency on audio speech during synthesis, allowing the synthesized talking head to provide additional supplementary visual information.

## 2.2 SPEECH-TO-SPEECH TRANSLATION.

The speech-to-speech translation (Jia et al., 2019; Lee et al., 2021; Huang et al., 2022) aims to translate speech from one language to the semantically consistent speech of other languages, and has great promise for applications in scenarios such as transnational meetings and online education. Earlier work (Lavie et al., 1997; Zhang et al., 2021) was based on a cascade model approach to speech-to-speech translation, where the source speech is first recognized as text, and then the translated text in the new language is synthesized into speech in the target language. However, the excessive number of cascading models results in extremely slow model inference, which is not suitable for online scenarios with high real-time requirements, and introduces additional cascading errors, and is also not suitable for languages (e.g., Minnan) and dialects without text. Some studies (Lee et al., 2021; Huang et al., 2022) employ a self-supervised learning model to represent audio speech as discrete units during training. They then proceed to train a speech-to-unit translation model (S2UT). This enables them to resynthesize the corresponding audio speech based on these discrete units, facilitating text-independent speech-to-speech translation.

However, the development of the talking head translation task (KR et al., 2019; NVIDIA, 2022; Waibel et al., 2022; Song et al., 2022; Waibel et al., 2023) is currently in its early stages. All of the schemes (KR et al., 2019; Song et al., 2022; Waibel et al., 2023) are based on a cascade model, from the speech of origin language, through ASR (Yu & Deng, 2016; Potamianos et al., 2004), NMT (Brown et al., 1990; Bahdanau et al., 2014), TTS (Klatt, 1987; Ren et al., 2019; 2020), and wav2lip (Prajwal et al., 2020) models to finally achieve the talking head translation. The lack of parallel visual corpus for the talking head translation has led existing work into training with such super-multiple cascade models. Furthermore, if the translation result of the talking head is too lengthy, it necessitates the reuse of the reference frame, leading to the jarring video transition. When applied in streaming scenarios, achieving isometric translation is also imperative. In this work, we present a system for direct talking head translation, TransFace, consisting of a speech-to-discrete-unit translation module (S2UT), and a unit-based audio-visual speech synthesizer (Unit2Lip). It realizes learning cross-linguistic mappings from parallel audio-speech corpus and overcomes the challenge of no parallel visual corpus. We also propose a bounded-duration-predictor that dynamically adjusts the duration of each unit, realizing isometric translation, and addressing jarring video transition.

## 3 TRANSFACE

The illustration of Direct Talking Head Translation system (TransFace) for audio-visual speech has been presented in Figure 1. (1) As shown in Section 3.1, we employ a self-supervised learning (SSL) model HuBERT (Hsu et al., 2021) that has been pre-trained on the audio speech corpus to derive discrete units of target audio speech. These units are then employed to train the Speech-To-Unit Translation model (S2UT). Furthermore, the SSL model is also adopted to obtain discrete units of audio speech from the audio-visual speech, which are used to train the unit-based synthesizer. (2) In Section 3.2, we introduce a speech-to-unit translation (S2UT) model to translate the source audio speech into the target units. (3) Subsequently, a unit-based audio-visual speech synthesizer, which

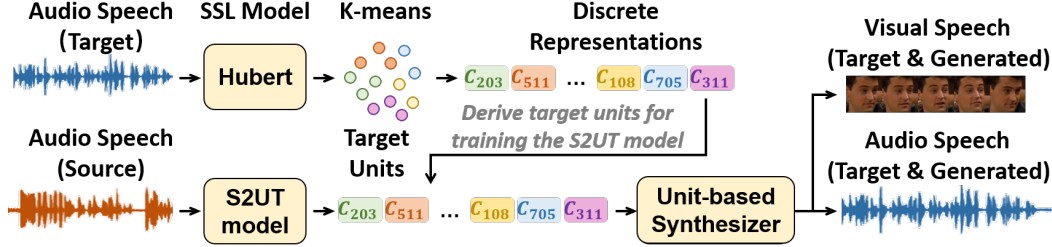

Figure 1: Illustration of the direct Talking Head Translation system for audio-visual speech. Both audio and visual speech can be taken as input or be generated. In this paper, we present a sample that takes audio speech as input and generates synchronized audio-visual speech. The unit-based synthesizer consists of two parallel synthesizers which can synthesis the corresponding audio speech and visual speech from the same units, respectively.

is trained separately on the target language audio-visual dataset, is applied to convert the translated units into target audio-visual speech, as detailed in the Section 4.

### 3.1 HuBERT AND DISCRETE UNITS

HuBERT (Hsu et al., 2021) is a self-supervised learning (SSL) model trained on unlabeled audio speeches, using an iterative approach that alternates between feature clustering and mask prediction. In each iteration, the discrete labels of the audio speech sequences are generated through feature clustering on intermediate representations (or Mel-frequency cepstral coefficient features for the first iteration), and these labels are then used to compute a BERT-like mask prediction loss. The target audio speech $A^{tgt} = \{a_1^{tgt}, \cdots, a_T^{tgt}\}$ is encoded into continuous units $Z^{tgt} = \{z_1^{tgt}, \cdots, z_T^{tgt}\}, z_i^{tgt} \in \{0, 1, \cdots, K-1\}$ at every 20-ms frame, where $a_i$ and $z_i$ are the $i$-th acoustic frame and its clustering unit, $T$ is the number of frames and $K$ is the number of cluster centers. We obtain the discrete units of target audio speech from the parallel speech-to-speech translation dataset (ie., LRS3-T) to train the S2UT model, and the discrete units of target language audio-visual speech from the audio-visual speech dataset (ie., LRS2) to train the unit-based audio-visual speech synthesizer.

### 3.2 SPEECH-TO-UNIT TRANSLATION MODEL

Denote $A^{src} = \{a_1^{src}, \cdots, a_T^{src}\}$ and $U^{tgt} = \{u_1^{tgt}, \cdots, u_N^{tgt}\}, u_i^{tgt} \in \{0, 1, \cdots, K-1\}$ as the source language audio and target language full *orig-unit* sequence discrete units. The *orig-unit* sequence is obtained by removing repeating units from the continuous units sequences $Z^{tgt}$, resulting a sequence of unique discrete units. The S2UT model autoregressively decode the source audio speech into the target probabilities: $p(u_t | \{u_i\}_{i=1}^{t-1}, A^{src}) = \texttt{S2UT}(A^{src})$. And, the S2UT model is trained with the cross-entropy loss :

$$L_{s2s} = -\sum_{t=1}^{N} \log p(u_u | \{u_i\}_{i=1}^{t-1}, A^{src}). \tag{1}$$

### 4 UNIT-BASED AUDIO-VISUAL SPEECH SYNTHESIZER

In this section, we present a unit-based audio-visual speech synthesizer, Unit2Lip, that effectively converts discrete units into synchronized audio and visual speech. Although the unit-based vocoder (Polyak et al., 2021; Kong et al., 2020), which transforms units into audio speech, has been extensively studied and applied, the synthesizer for audio-visual speech encounters several additional challenges: (1) ensuring the synchronization of audio and visual speech, and (2) preserving the output video length identical to the source video length.

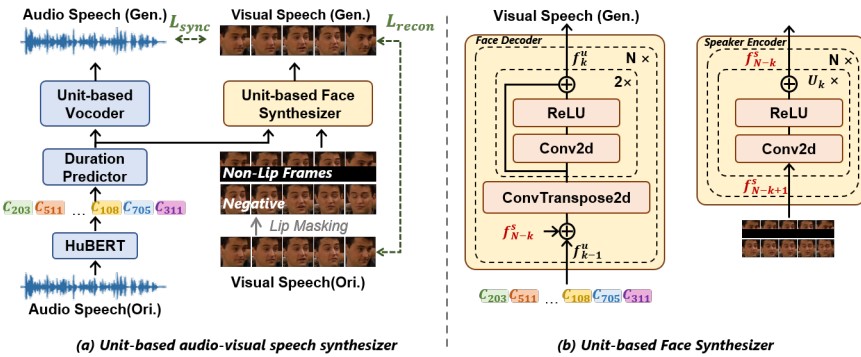

Figure 2: llustration of the *(a) unit-based audio-visual speech synthesizer* and the *(b) unit-based face synthesizer*. The unit-based audio-visual speech synthesizer includes a pre-trained **unit-based vocoder** for audio speech synthesis and a **unit-based face synthesizer** for visual speech synthesis. **Speaker Encoder** is a stack of residual convolutional layers that encode the non-lip frames (target faces with lower-half masked) and negative frames (random reference faces). **Face Decoder** is a stack of convolutional layers, along with transpose convolutions for upsampling.

### 4.1 DISCRETE UNITS OF AUDIO-VISUAL SPEECH STILL BASED ON ACOUSTIC FEATURES.

Although there is already audio-visual speech self-supervised learning (Shi et al., 2022) which can represent audio-visual speech as corresponding discrete units, it is not able to extract the discrete units corresponding to audio-only speech. In contrast, since audio speech and visual speech are temporally aligned in parallel, discrete units from audio-only speech can also be used directly for audio-visual speech representation. In this work, for S2UT models to be trained using parallel audio speech corpus without visual speech, we utilize mHuBERT (Lee et al., 2021), trained on multilingual and large-scale audio-only speech, to generate clustering units based on only acoustic features in audio-visual speech.

### 4.2 BOUNDED DURATION PREDICTOR

The S2UT model decodes the discrete unit sequence in an autoregressive manner, but it does not include the duration information for each unit. The duration predictor is a two-layer 1D convolutional network with ReLU activation, each followed by layer normalization and dropout layer, and an additional linear layer that outputs a scalar, which precisely predicts the duration of each unit. To ensure that the generated audio-visual speech has the same duration as the original audio-visual speech (isometric translation), we propose a bounded duration predictor that bounds the length of input unit sequence. The predicted duration sequence for each unit using the duration predictor can be represented as: $D = \{D_1, \cdots, D_N\}$, where $D_i$ is the predicted duration of $i$-th unit. Subsequently, after normalizing the entire duration sequence, it is multiplied by the target sequence length $T$ (i.e., the length of the original audio sequence) to impose length constraints: $D'_i = \frac{D_i * T}{\sum_{i=1}^{N} D}$. We select $T$ units from highest to lowest for the generation task. [Revised Part 1] For instance, when $U = \{u_1, u_2, u_3, u_4\}$, $D' = \{2.2, 1.8, 2.3, 2.7\}$ and $T = 10$, the weight of each frame can be denoted as $U' = \{1.0u_1, 1.0u_1, 0.2u_1, 1.0u_2, 0.8u_2, 1.0u_3, 1.0u_3, 0.3u_3, 1.0u_4, 1.0u_4, 0.7u_4\}$, where the weight of $0.2u_1$ is only 0.2. And after enough 10 frames have been selected in order of highest to lowest weight $(1.0u_1, 1.0u_1, 1.0u_2, 1.0u_3, 1.0u_3, 1.0u_4, 1.0u_4, 0.8u_2, 0.7u_4, 0.3u_3, \cancel{0.2u_1})$, $0.2u_1$ is discarded. The sequence of input discrete units can be represented as $U' = \{u_1, u_1, u_2, u_2, u_3, u_3, u_3, u_4, u_4, u_4\}$. In the Appendix A, we provide a detailed pseudo-code for the bounded duration predictor.

### 4.3 UNIT-BASED AUDIO-VISUAL SPEECH SYNTHESIZER

[Revised Part 2] As shown in Figure 2, the unit-based audio-visual speech synthesizer is utilized to synthesis the audio-visual speech as the talking head video from a sequence of discrete units, which can be considered to be a modified version of Wav2Lip (Prajwal et al., 2020). For audio

speech, we adopt the pretrained unit-based vocoder (Polyak et al., 2021) and leave it unchanged. The Unit-Based Face Synthesizer architecture (Figure 2.b) consists of a generator $G$ and a discriminator $D$. The generator $G$ consists of three blocks: (i) a set of looked-up tables (LUT) that embed the representation $f^u$ of discrete units; (ii) `Speaker Encoder` to extract speaker representation $f^s$ from random reference frames and pose-prior frames (target-face with lower-half masked); and (iii) `Face Decoder` to upsample the encoded representation to match the input sample rate. The sequence of discrete units $U = \{u_1, \cdots, u_N\}$ is transformed into a continuous representation through $f_i^u = \text{LUT}(u_i)$. The speech representation $f^u = \{f_1^u, \cdots, f_N^u\}$ is then up-sampled and concatenated with the speaker representation $f^s = \{f_1^s, \cdots, f_N^s\}$. The discriminator $D$ consists of a stack of convolutional blocks, and is trained in turn with generator $G$.

## 4.4 TRAINING LOSS TERMS

**GAN Loss.** For generator $G$ and discriminator $D$, the training objectives are as follows, where $I_{gen}$ are the frames of the generated visual speech and $I_{reak}$ are the real frames:

$$\mathcal{L}_G = \mathbb{E}_{x \sim I_{gen}} \log(1 - D(x)); \qquad \mathcal{L}_D = \mathbb{E}_{x \sim I_{real}} \log(1 - D(x)) + \mathbb{E}_{x \sim I_{gen}} \log(1 - D(x)). \quad (2)$$

**Lip Reconstruction Loss.** In addition to the GAN objective, we employ the lip-reconstruction loss to improve the efficiency of the generator and the realism of the generated lips. Following the previous work, reconstruction loss is applied to assist the generator training by limiting the L1 distance between the generated lips and the real ones to ensure the temporal realism of the generated lips. The lip reconstruction loss could be defined as:

$$\mathcal{L}_{Lip} = \frac{1}{N} \sum_{i=1}^{N} (\|I_{real} - I_{gen}\|_1) \quad (3)$$

**Synchronicity Loss.** [Revised Part 3] The synchronization expert has proven to be a valuable way to improve the synchronization of the generated talking head with the audio speech (Prajwal et al., 2020). For the unit-based audio-visual speech synthesizer, we minimize the distance (Chung & Zisserman, 2017) between the synthesized audio speech and visual speech to improve the synchronization of the generated video, where $a$ and $v$ are the representation of audio and visual speeches and $P_{sync}$ is the similarity of $a$ and $v$:

$$P_{sync} = \frac{v \cdot a}{\max(\|v\|_2 \cdot \|a\|_2)}; \qquad \mathcal{L}_{sync} = \frac{1}{N} \sum_{i=1}^{N} -\log(P_{sync}^i). \quad (4)$$

The overall modal is trained with the $\mathcal{L} = (1 - \lambda_{sync} - \lambda_{gen})\mathcal{L}_{Lip} + \lambda_{sync}\mathcal{L}_{sync}^{gen} + \lambda_{gen}\mathcal{L}_G$, where $\lambda_{sync} = 0.03$ and $\lambda_{gen} = 0.07$ as the the setting of (Prajwal et al., 2020).

## 5 EXPERIMENTS

### 5.1 EXPERIMENTAL SETUP

**Dataset.** For the Audio-Visual Speech Resynthesis task, we adopt the setup from the previous works (Prajwal et al., 2020) on talking head generation, training on the widely utilized LRS2 dataset (Afouras et al., 2018a). In the Speech(A)-To-Speech(AV) Translation task, we conduct experiments on the LRS3-T dataset (Huang et al., 2023). This dataset serves as a parallel translation corpus, featuring English audio-visual speech alongside audio-speech from other languages (Spanish and French). It is constructed based on the LRS3 dataset (Afouras et al., 2018b) using the S2ST dataset construction pipeline as outlined in CVSS-T (Jia et al., 2022).

**Evaluation Metrics.** [Revised Part 4] In the audio-visual speech resynthesis task, we employ various metrics. This includes utilizing FID (Heusel et al., 2017) to gauge image similarity as (Prajwal et al., 2020), employing LSE-C and LSE-D for measuring audio-visual synchronization (Chung & Zisserman, 2017), and assessing mean opinion score(MOS) for both synchronization and image quality. For the translation task, we utilize AVSR (Shi et al., 2022) to recognize the textual content of the translated results and measure the translation quality with SACREBLEU (Post,

Table 1: Comparison of the speed and quality among different talking head synthesis methods. Audio (ori.) and Audio (gen.) respectively represent the synchronization of the synthesized talking-head with the source audio speech and the synchronization of the synthesized audio-visual speech, respectively. ($\times$ Rate) in parentheses represents the speed compared to real-time.

| Method | Audio(ori.) | | Audio(gen.) | | FID↓ | Speed(FPS)↑ | MOS↑ |
|--------|--------|--------|--------|--------|------|-------------|------|
| | LSE-C↑ | LSE-D↓ | LSE-C↑ | LSE-D↓ | | | |
| *Real Audio Visual Speeches* | | | | | | | |
| Audio(GT)+Wav2lip | 6.498 | 7.143 | / | / | 5.08 | / | 4.23±0.12 |
| Video(GT) | 6.221 | 7.407 | / | / | / | / | 4.15±0.25 |
| *Resynthesized Audio Visual Speeches* | | | | | | | |
| U2S+LipGAN | 2.875 | 11.431 | 3.057 | 10.964 | 11.91 | 671.49(x26.86) | 2.64±0.23 |
| U2S+Wav2Lip | 5.742 | 7.958 | 6.298 | 7.305 | 5.52 | 544.37(x21.78) | 3.92±0.22 |
| Unit2Lip(ours) | **7.343** | **7.193** | **7.280** | **7.097** | **5.14** | **2369.66(x94.79)** | **3.98±0.24** |

2018). Simultaneously, we gather Mean Opinion Scores (MOS) to assess the translation quality, image quality, synchronization, and overall sensation of the translated results. [Revised Part 5] The detailed evaluation process of MOS are presented in Appendix B.3. Furthermore, we introduce two additional metrics, length ratio (LR) and length compliance (LC), in the form of Isometric translation (Antonios et al., 2022). LR denotes the ratio of the length of the prediction result to the length of the original, and LC@k denotes the acceptance of samples whose predicted length is within $\pm k\%$ of the original length.

**Implementation details.** Following the unit-based S2ST approach as outlined in (Lee et al., 2021), we applied the k-means algorithm to cluster the representations generated by the normalized mHu-BERT (Huang et al., 2022) into a vocabulary of 1000 units, establishing a discrete unit representation for audio-visual speech. For the Audio-Visual Speech Resynthesis task, we followed the previous work by training on the 29-hour training set from LRS2 (Prajwal et al., 2020) and leave the unit-based HiFi-GAN vocoder (Polyak et al., 2021) unchanged. Our training process encompassed 300K steps on a single 3090 GPU. As for S2ST task, we performed 200K steps of training on a single 3090 GPU, consistent with previous methods (Lee et al., 2021). In the inference process, we employ the S2UT model to translate the audio speech from the source language into a sequence of discrete units in the target language. These discrete units are subsequently synthesized into the corresponding target language audio-visual speech with Unit2Lip. More experimental details are shown in Appendix B.

## 5.2 WHY UNIT-BASED INSTEAD OF MEL-BASED FACE SYNTHESIZER?

In Table 1, we make a comparison of the performance of the unit-based audio-visual speech synthesizer (Unit2Lip) and the mel-based Talking Head Generation methods (LipGAN (KR et al., 2019) and Wav2Lip (Prajwal et al., 2020)) in terms of speed and quality. Note that all audio-visual speech resynthesis here begins with discrete units. Therefore, in the traditional mel-based talking head generation method, a unit-based vocoder (Polyak et al., 2021) (U2S) is needed to generate audio speech first. Experiments demonstrate that the unit-based audio-visual speech resynthesis method (Unit2Lip) is remarkable for the following aspects: (1) **Acoustic Retention.** As a single-stage model, the Unit2Lip can synthesize the corresponding visual speech directly from discrete units, avoiding the cascading errors introduced by intermediate processes. Compared to the two-stage models (e.g. u2s+wav2lip), the single-stage model (ie., Unit2Lip) is more synchronized with the original audio speech (LSE-C deteriorated by 1.601 and LSE-D by 0.765). At the same time, the Unit2Lip is also comparable to the Wav2Lip scheme in terms of synchronization with the audio(gen.), demonstrating that discrete units can adequately express the corresponding audio content.(2) **Rapid Processing.** Unit2Lip enables parallel synthesis of audio speech and visual speech (Unit-To-Audio & Lip), significantly enhancing the inference speed compared to the serial synthesis mode (Unit-To-Audio-To-Lip). Additionally, the introduction of codebook further reduces the computation time of the audio encoder. Experimental results demonstrate that Unit2Lip achieves

Table 2: **BLEU** scores (↑) and mean opinion scores (**MOS** ↑) of Speech(A)-To-Speech(AV) Translation on Es-En and Fr-En of LRS3-T. **Textual Dataset** denotes an additional textual dataset used, **SRC** denotes the text in the source language and **TGT** denotes the text in the target language. **NMT**: Neural Machine Translation, **ST**: Speech Translation, **ASR**: Automatic Speech Recognition, **TTS**: Text-to-Speech.

| ID | Method | Textual Dataset | | Es-En | | Fr-En | |
|---|---|---|---|---|---|---|---|
| | | **SRC** | **TGT** | **BLEU↑** | **MOS↑** | **BLEU↑** | **MOS↑** |
| | *Ground Truth:* | | | | | | |
| **0** | Video | ✗ | ✗ | 97.04 | - | 97.04 | - |
| **1** | Audio+Wav2Lip | ✗ | ✗ | 87.73 | - | 87.73 | - |
| | *Generated From Grount Truth Units:* | | | | | | |
| **2** | Unit+U2S+Wav2Lip | ✗ | ✗ | 81.03 | 4.22±0.11 | 80.71 | 4.18±0.10 |
| **3** | Unit+Unit2Lip(ours) | ✗ | ✗ | **83.98** | **4.35±0.07** | **83.89** | **4.32±0.08** |
| | *Cascaded Models:* | | | | | | |
| **4** | ST+TTS+Wav2Lip | ST | TTS | 51.20 | 3.76±0.05 | 42.60 | 3.68±0.11 |
| **5** | ASR+NMT+TTS+Wav2Lip | ASR+NMT | NMT+TTS | 64.09 | 4.16±0.08 | 52.34 | 4.12±0.70 |
| | *Direct System with High Resource:* | | | | | | |
| **6** | Translatotron2+Wav2Lip | ✗ | ✗ | 42.31 | 3.34±0.12 | 36.44 | 3.22±0.12 |
| **7** | S2ST+Wav2Lip | ✗ | ✗ | 60.93 | 3.99±0.07 | 45.17 | 3.76±0.06 |
| **8** | TransFace(ours)+bounded | ✗ | ✗ | 61.06 | **4.25±0.07** | 46.78 | **4.21±0.10** |
| **9** | TransFace(ours) | ✗ | ✗ | **61.93** | 4.12±0.06 | **47.55** | 3.88±0.07 |

Table 3: Comparison of translation quality (**BLEU** and **MOS**) as well as length metrics (**LR** for length ratio and **LC** for length compliance) across various methods for Spanish to English (Es-En) translation on LRS3-T. *Early Stop*: Finishing the synthesis according to the frame length. *Bounded*: Synthesising with the bounded duration predictor.

| Method | BLEU↑ | MOS↑ | LR | LC@5↑ | LC@10↑ | LC@20↑ |
|---|---|---|---|---|---|---|
| ASR+NMT+TTS+Wav2Lip | **64.09** | 4.16±0.08 | +0.082(1.082) | 12.38 | 32.64 | 53.88 |
| Translatorn2+Wav2Lip | 42.31 | 3.34±0.12 | +0.067(1.067) | 16.87 | 34.23 | 48.10 |
| S2ST+Wav2Lip | 60.93 | 3.99±0.07 | +0.055(1.055) | 15.80 | 31.87 | 57.46 |
| TransFace+Early Stop | 52.76 | 4.02±0.08 | -0.051(0.949) | 77.63 | 83.79 | 93.76 |
| TransFace+bounded | 61.06 | **4.25±0.07** | **0.000(1.000)** | **100.00** | **100.00** | **100.00** |

an impressive speedup of × 4.35 times compared to other methods. **(3) High Fidelity.** Since the discrete units obtained from pre-training effectively distinguish different phonemes, Unit2Lip can easily learn the mapping from these discrete units to the shape of the target lips, enabling the synthesis of high-fidelity lip movements that are comparable to mel-based methods. We present some synthesized talking head samples in different languages in Appendix C.1. **(4) Length Regulation.** Talking head synthesis, typically accomplished by adjusting the lip movements of reference frames, can result in abrupt video transitions if these frames are repeated. By utilizing discrete units, we can control the length of the synthesized talking head video as desired. For example, in Section 5.3, we demonstrate the synthesis of an translated talking head, matching the length of the reference frames.

### 5.3 DIRECT SPEECH(A) TO SPEECH(AV) TRANSLATION.

We present the results of the Speech(A)-To-Speech(AV) Translation experiments conducted on LRS3-T, as outlined in Table 2, yielding the following observations: **(1) Cascade System vs. Direct System (#5 vs. #9).** Despite the cascade system utilizing more textual data for training, there is minimal disparity in translation quality between the two (with only a 2.16 difference in BLEU score, from 64.09 to 61.93). In contrast, the unit-based approach (#8), which achieves the length regulation and ensures non-repeat of video reference frames, notably enhances user acceptance (MOS, improved from 4.16 to 4.25). **(2) Mel-Based vs. Unit-Based (#2, 7 vs. #3, 9).** The visual speech

produced by the mel-based method relies entirely on synthesized audio speech. In contrast, the unit-based approach handles the synthesis of audio and visual speech independently, providing additional supplementary visual information (e.g., visually distinguishable phonemes /ae/ and /ai/) to the audio speech. This leads to a improvement in translation results, with the BLEU score rising from 60.93 to 61.93. **(3) S2Spec vs S2U (#6 vs. #9).** With the introduction of discrete units, we convert continuous target speech into discrete semantic tokens as training targets, so that the S2ST task can be trained with well-defined objective as an autoregressive task. Comparing with the reconstruction-method, from Speech to Melspectrograms (S2Spec, #6), the autoregressive method effectively reduces the difficulty of the training, and improves the translation quality from 42.31 to 61.93 improved by 19.62 BLEU. **(4) Unit-based Isometric Translation.** Isometric translation can effectively improve the user acceptance in the Talking Head Translation. We show the comparison of the length metrics and translation quality of the different synthesis methods in Table 3. We find that the unit-based method `TransFace+bound` achieves the 100% isometric translation without degrading the quality of the generated translated talking head video. Since it is fully ideographic and solves the jarring video transitions, user acceptance is significantly improved (MOS from 4.12 to 4.25). To further qualify the visual speech of our generated talking heads, we conducted additional ablation experiments focusing on visual speech, the results of which are detailed in Appendix C.2. These experiments demonstrate that our TransFace is able to generate talking heads that correspond to the translated content, while also having semantic complementarities with the audio speech.

## 5.4 CASE STUDY

We present some translation results in Table 4, where the results of TransFace consistently maintain a high semantic consistency with the target results. This substantiates the model can comprehend discrete unit sequences and establish cross-language mappings between them. Furthermore, upon reviewing the translated talking head videos on the demo page, we observe a noticeable frame jitter phenomenon when the bounded-duration-predictor method is not employed. This significantly impacts the quality of the talking head videos. In contrast, the TransFace+bounded method achieves a remarkable 100% isometric translation while ensuring translation quality. Qualitative experiments demonstrate that our TransFace+bounded framework exhibits translation quality on par with cascade models, while also delivering a higher level of realism.

Table 4: Comparison of translation quality among different methods. ~~Red Strikeout Words~~: mistranslated words with opposite meaning, Blue Words: mistranslated words with similar meaning, Gray Words: the absent words.

|        |                    |                                                                  |
|--------|--------------------|------------------------------------------------------------------|
|        | Source(Es)         | Así que el agua era algo que me asustaba para empezar.           |
|        | Target(En)         | so water was something that scared me to begin with.             |
|        | ASR+NMT+TTS+wav2lip | so water was something that scared me in the beginning to ....   |
| **Es-En** | ST+TTS+wav2lip   | so water was something that scared me to begin with.             |
|        | S2ST+Wav2Lip       | so water was something that was scaring me to start begin with.  |
|        | TransFace          | so water was something that was scaring me to start begin with.  |
|        | TransFace+bounded  | so water was something that was scaring me to start begin with.  |

## 6 CONCLUSION

Talking head translation aims to translate speech from one language to audio-visual speech (i.e., talking head video) in other languages, which is widely used in online conferencing, online healthcare and online education. However, all current methods rely on cascading multiple models, severely hampering inference speed and rendering them unsuitable for online streaming applications. In this paper, we introduce TransFace, a direct talking head translation model comprising a speech-to-discrete-unit translator and a unit-based audio-visual speech synthesizer, Unit2Lip. This framework facilitates parallel synthesis of audio-visual speech, significantly enhancing the speed of generating translated talking heads. Additionally, we propose a bounded duration predictor to ensure consistent translation without compromising quality, addressing the challenge of abrupt video variations. Experiments demonstrate that TransFace achieves the state-of-the-art BLEU score, user acceptance, and 100% isometric translation on LRS3-T.

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

# A  THE DETAILS OF THE BOUNDED DURATION PREDICTOR.

## A.1  ALGORITHMIC DETAILS

[Revised Part 6] In Algorithm 1, we present the details of the bounded-duration-predictor algorithm:

1. After normalization, the duration $\mathbf{D}$ can be computed as the allocated duration $\mathbf{D}'$ corresponding to each token based on the target length $\mathbf{T}$:

$$\mathbf{D}' = \texttt{Normalize}(\mathbf{D}) \times \mathbf{T}. \tag{5}$$

2. Following the rounding method, it is converted to an integer predicted duration $\mathbf{PRED}$:

$$\mathbf{PRED} = \texttt{Clamp}(\texttt{Round}(\mathbf{D}'), min = 1). \tag{6}$$

3. Calculate the difference $\mathbf{DIFF}$ between the predicted duration $\mathbf{PRED}$ and the allocated duration $\mathbf{D}'$ for each token:

$$\mathbf{DIFF} = \mathbf{D}' - \mathbf{PRED}. \tag{7}$$

4. Determine whether the predicted duration $\mathbf{PRED}$ still needs adjustment in the number of frames. If an increase is required, select the highest-weighted difference $\mathbf{DIFF}$ from the sequence for its duration+1. Conversely, if a decrease is needed, select the lowest-weighted difference $\mathbf{DIFF}$ for its duration-1.

---

**Input :**
    T: The desired length of the translation result.
    D: The predicted duration sequence for each unit.
**Output:**
    OUT: The duration of each discrete cell after length regulation.

```
//Step1:
```
$\mathbf{D}' = \texttt{Normalize}(\mathbf{D}) \times \mathbf{T}$;

```
//Step2:
```
$\mathbf{PRED} = \texttt{Clamp}(\texttt{Round}(\mathbf{D}'), min = 1)$ ;

```
//Step3:
```
$\mathbf{DIFF} = \mathbf{D}' - \mathbf{PRED}$ ;

```
//Step4:
```
$\mathbf{ADD} = \texttt{Zeroes}()$ ;
**if** $\texttt{Sum}(\mathbf{PRED}) > \mathbf{T}$ **then**
    $\mathbf{INDEX} = \texttt{TopK}(-\mathbf{DIFF}, k = \texttt{Sum}(\mathbf{PRED}) - \mathbf{T})$ ;
    $\mathbf{ADD}[\mathbf{INDEX}] = -1$ ;
**else**
    $\mathbf{INDEX} = \texttt{TopK}(\mathbf{DIFF}, k = \mathbf{T} - \texttt{Sum}(\mathbf{PRED}))$ ;
    $\mathbf{ADD}[\mathbf{INDEX}] = 1$ ;
**end**
$\mathbf{OUT} = \mathbf{PRED} + \mathbf{ADD}$

**Algorithm 1:** Pseudo-code for bounded-duration-predictor implementation details.

---

## A.2  IMPLEMENTATION SAMPLE

[Revised Part 7] Let's revisit the previous example in section 4.2 for illustration, when $U = \{u1, u2, u3, u4\}$, $D' = \{2.2, 1.8, 2.3, 2.7\}$ and $T = 10$:

- Step1: $\mathbf{D}' = [2.2, 1.8, 2.3, 2.7], \mathbf{T} = 10$.
- Step2: $\mathbf{PRED} = [2, 2, 2, 3]$.
- Step3: $\mathbf{DIFF} = [0.2, -0.2, 0.3, -0.3]$.
- Step4: Since $\texttt{SUM}(\mathbf{PRED}) = 9 < 10$, for the largest $\mathbf{T} - \texttt{SUM}(\mathbf{PRED}) = 1$ corresponding token in $\mathbf{DIFF}$, its duration+1, resulting in $\mathbf{OUT} = [2, 2, 2+1, 3] = [2, 2, 3, 3]$.

The sequence of discrete units can be represented as $U' = \{u_1, u_1, u_2, u_2, u_3, u_3, u_3, u_4, u_4, u_4\}$.

| Method | Translation | Image | SYNC | Overall | Mean |
|---|---|---|---|---|---|
| ST+TTS+Wav2Lip | 3.78±0.05 | 4.03±0.08 | 3.66±0.04 | 3.57±0.03 | 3.76±0.05 |
| ASR+NMT+TTS+Wav2Lip | **4.23±0.06** | 4.11±0.07 | 4.12±0.08 | 4.18±0.11 | 4.16±0.08 |
| Translatotron2+Wav2Lip | 2.79±0.09 | 3.98±0.12 | 3.68±0.07 | 2.91±0.20 | 3.34±0.12 |
| S2ST+Wav2Lip | 4.03±0.08 | 4.05±0.04 | 4.02±0.09 | 3.86±0.07 | 3.99±0.07 |
| TransFace(ours) | 4.19±0.08 | 4.08±0.07 | **4.28±0.04** | 3.93±0.05 | 4.12±0.06 |
| TransFace(ours)+bounded | 4.17±0.06 | **4.16±0.06** | **4.28±0.05** | **4.39±0.11** | **4.25±0.07** |

Table 5: The detailed MOS (Mean Opinion Score) results for talking head translation. Each dimension is scored individually on a scale of 1 (lowest) to 5 (highest). **Translation**: translation quality, **Image**: image quality, **SYNC**: Synchronization, **Overall**: overall sensation.

| Method | Image quality | Synchronization | Mean |
|---|---|---|---|
| Audio(GT)+Wav2lip | 4.18±0.32 | 4.12±0.18 | 4.15±0.25 |
| Video(GT) | **4.33±0.13** | **4.13±0.11** | **4.23±0.12** |
| U2S+LipGAN | 2.89±0.25 | 2.39±0.21 | 2.64±0.23 |
| U2S+Wav2Lip | **4.01±0.20** | 3.93±0.24 | 3.92±0.22 |
| Unit2Lip(ours) | 3.95±0.24 | **4.01±0.24** | **3.98±0.24** |

Table 6: The detailed MOS (Mean Opinion Score) results for unit-based talking head generation. Each dimension is scored individually on a scale of 1 (lowest) to 5 (highest).

## B MORE IMPLEMENTATION DETAILS

### B.1 DATA PREPROCESSING.

For visual speech, we extract the facial region from the video for Unit2Lip model training. As in prior research (Prajwal et al., 2020; Shi et al., 2022), we use dlib (King, 2009) to detect 68 facial keypoints, and then isolate a 96x96 region-of-interest (ROI) video segment centered around the face. For the source language audio speech, we extract 80-dimensional mel-filterbank features at 20-ms intervals as input. Regarding the target language audio speech, we apply the k-means algorithm to cluster the representations provided by the well-tuned mhubert into 1000 discrete units for training purposes.

### B.2 MODEL CONFIGURATION AND TRAINING DETAILS.

We adopted the same S2UT model architecture as (Lee et al., 2021), employing 8 attention heads and a embedding size of 512. We select one discrete unit every 20ms to synthesize one audio frame and every 40ms to synthesize one visual frame. The audio-speech vocoder utilizes the unit-based vocoder pre-trained in (Lee et al., 2021), whereas the decoder of the visual-speech synthesizer employs the same architecture as Wav2lip (Prajwal et al., 2020). In the inference process, since the audio speech in language X lacks a corresponding visual speech as reference frames, we utilize the visual speech of the English videos as reference frames for synthesizing the translated talking head. All the cascade models we utilized are publicly available pre-trained systems in Fairseq (Ott et al., 2019). For instance, we employed MMT to convert text from other languages to English, and the FastSpeech2 model to transform text into corresponding audio speech.

### B.3 THE DETAILED EVALUATION PROCESS OF MOS

[Revised Part 8] Our comprehensive MOS scoring process for talking head translation tasks involves gathering scores across four dimensions: translation quality, image quality, synchronization, and overall sensation. For the unit-based talking head generation, we streamline the evaluation to two dimensions: image quality and synchronization. Translation quality assesses the consistency of the translated content with the original sentence, while image quality evaluates the presence of artifacts in the generated image. Synchronization measures the coherence of audio and visual speech,

and overall sensation indicates the evaluation of the video's authenticity. Each sample is randomly scrambled and presented to 15 participants for scoring. A composite MOS is then calculated by averaging the scores for the corresponding dimensions, with each dimension scored individually on a scale of 1 (lowest) to 5 (highest). Here, we present the MOS (Mean Opinion Score) results for both talking head translation in Table 5 and unit-based talking head generation in Table 6.

## C   MORE EXPERIMENTS

### C.1   SAMPLES OF UNIT-BASED TALKING HEAD GENERATION

We present samples of our unit-based talking head generation method (Unit2Lip). For each discrete unit, we equidistantly selected 6 pairs of corresponding original and generated Talking Head video frames. Notably, the lip shapes between each pair of images are highly consistent, suggesting that the model is adept at reconstructing the lip shapes associated with discrete units while preserving more of the original video information. Furthermore, we conduct experiments in French and the lip shape consistency is also well-maintained, demonstrating that the Unit2Lip can be generalized across languages, not only to English but also to different languages.

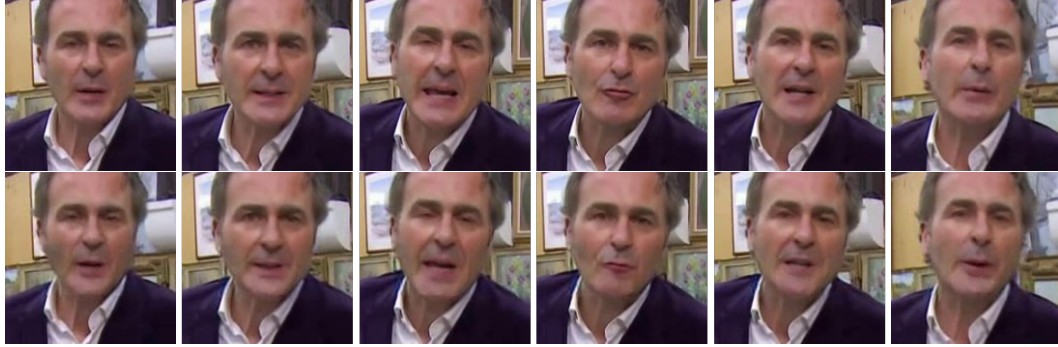

(a) Resynthesis sample for English. The top row is the original, the bottom row is the synthesized one.

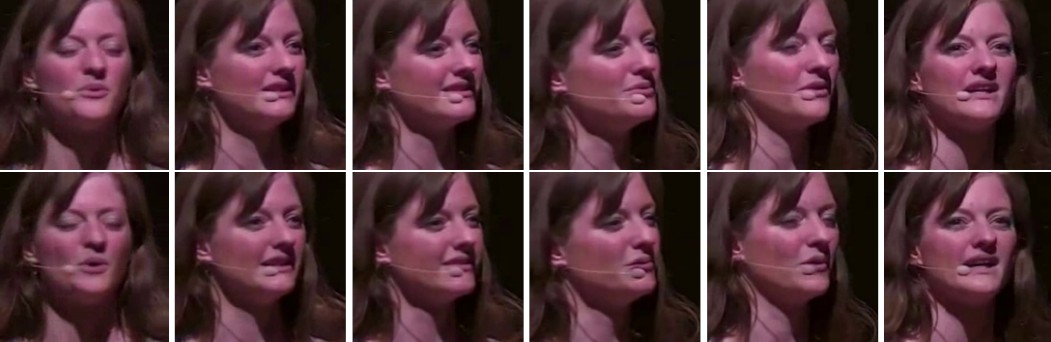

(b) Resynthesis sample for French. The top row is the original, the bottom row is the synthesized one.

Figure 3: Quantitative comparison of audio and visual speech resynthesized from discrete units on English and French. More experimental results are available on the demo page.

### C.2   CAN THE TALKING HEAD REPRESENT THE CORRESPONDING CONTENT?

In this paper, due to the relatively lower discriminability of visual speech compared to audio speech, lip reading (visual speech recognition) yields significantly lower recognition accuracy. Therefore, most experiments in this paper employ audio-visual speech recognition (AVSR) results for computing BLEU scores. In this subsection, we conduct additional ablation experiments focused on visual speech to investigate its effectiveness in representing the corresponding content. In Table 7, we show the BLEU comparisons for different modalities of speech in different methods. **(1) The talking head can effectively convey the corresponding content.** Comparing with random frames

Table 7: BLEU scores of different methods with distinct modality speech on Es-En and Fr-En of LRS3-T. *Random Frame*: Randomly scrambled video that does not express any information.

|  | Method | Modality | Es-En | Fr-En |
|---|---|---|---|---|
| **1** | Audio(GT)+wav2lip | Visual-Only | 13.23 | 13.23 |
| **2** | Audio(GT)+wav2lip | Audio-Visual | 87.73 | 87.73 |
| **3** | Random frame | Visual-Only | 0.18 | 0.68 |
| **4** | S2ST+wav2lip | Visual-Only | 8.59 | **7.56** |
| **5** | TransFace | Visual-Only | **8.62** | 7.53 |
| **6** | TransFace | Audio-Only | 60.76 | 46.89 |
| **7** | S2ST+Wav2Lip | Audio-Visual | 60.93 | 45.17 |
| **8** | TransFace | Audio-Visual | **61.93** | **47.55** |

results (#3), the BLEU of the translated video (#5) is improved by 8.46, which proves that it contains some useful information and is not directly random synthesis. Meanwhile, compared with the translated video synthesized by wav2lip (#4), the results of TransFace and S2ST+Wav2Lip are basically the same (8.59 vs. 8.62), which indicates that the synthesized result of our method is basically consistent with that of wav2lip. **(2) The talking head can provide complementary information to audio speech.** Comparing the TransFace results for audio-only (#6) and audio-visual (#8), we can observe a further improvement in the BLEU of translation (from 60.76 to 61.93) with the additional introduction of talking head. This indicates that the talking head, synthesized directly from the discrete units, contains valuable supplementary information to audio speech, further enriching the speech content.

| Method | En | | Es | |
|---|---|---|---|---|
| | NMI(↑) | Purity(↑) | NMI(↑) | Purity(↑) |
| AV-HuBert | **43.7** | **65.8** | 12.8 | 9.1 |
| m-HuBert | 42.6 | 65.1 | **41.7** | **63.2** |

Table 8: Comparison of clustering effects of m-HuBert and AV-HuBert on En and Es.

### C.3    WHY M-HUBERT INSTEAD OF AV-HUBERT?

[Revised Part 9] Among various discrete unit schemes (encodec (Défossez et al., 2022), av-hubert(Shi et al., 2022), hubert(Hsu et al., 2021), mhubert(Lee et al., 2021), etc.), only mhubert is publicly available and widely used as a multilingual discrete unit extractor. Hence, we chose mhubert to extract discrete unit representations. As demonstrated in other research (Le et al., 2023), if we want the model to encode languages other than English, it must be pre-trained on the speech of these languages.

Here, we also present a comparison of the clustering effect (Shi et al., 2022) of avhubert and mhubert on different languages in Table 8. Notably, there is no difference in performance between the two in English, but their effectiveness varies significantly in other languages. Furthermore, in this paper, we showcase that discrete units based on acoustic-only feature can also be effectively utilized for visual speech synthesis.

### C.4    MORE TRANSLATION RESULTS

[Revised Part 10] In addition to the result of Es-En in Table 4, we further show the translation results of Fr-En in Table 9 to visualize the translation performance of our model on different language pairs. The results indicate that our approach consistently delivers high-quality translation performance across various languages, including Es-En and Fr-En. Additional translation results for other language pairs (En-Es and En-Fr) are available on the demo page.

Table 9: Comparison of translation quality on Fr-En among different methods. ~~Red Strikeout Words~~: mistranslated words with opposite meaning, Blue Words: mistranslated words with similar meaning, Gray Words: the absent words.

| | | |
|---|---|---|
| | Source(Fr) | No fuimos considerados la cosa real. |
| | Target(En) | we weren't considered the real thing. |
| | ASR+NMT+TTS+wav2lip | we were not weren't considered the real thing. |
| **Fr-En** | ST+TTS+wav2lip | we were not weren't considered the real ~~ones~~ thing. |
| | S2ST+Wav2Lip | we were not weren't considered ~~as~~ the real thing. |
| | TransFace | we were not weren't considered ~~as~~ the real thing. |
| | TransFace+bounded | we were not weren't considered ~~as~~ the real thing. |

## D  LIMITATIONS AND ETHICAL DISCUSSIONS.

### D.1  ISOMETRIC TRANSLATION IS A FUNDAMENTAL REQUIREMENT FOR TALKING HEAD TRANSLATION.

[Revised Part 11] In contrast to speech translation, talking head translation encounters a relatively fixed limit on video frame length, especially evident when dubbing a translated movie. In such instances, the voice actor for translated movies must synchronize the translation to match the original video's duration. The absence of a duration-bounded module in the video results in noticeable frame skipping, leading to a significant loss of realism (refer to the demo page for results without the bounded-duration predictor). This limitation renders the approach unsuitable for professional scenarios like online meetings and movie translating, where the number of generated video frames must align with the original reference video. **The introduction of the `Bounded-Duration-Predictor` becomes imperative in such cases, despite tradeoffs in other factors, as it effectively satisfies the fundamental requirements of talking head translation.** We also acknowledge this approach may cause excessive speedups and slow reads, we plan to address this concern in our future work. Specifically, we intend to investigate the vocabulary length of the generated content to further enhance the realism and authenticity of the translated videos.

### D.2  WHY ONLY COMPARE BLEU IN X-EN?

In this work, we only compare BLEU scores for X-En translation results. This is due to the scarcity of current audio/video speech datasets in languages other than English, as well as the notably poorer performance in audio/video speech recognition for languages other than English. These factors contribute to a lack of convincing and credible results in those cases. Nonetheless, we still showcase the relevant translation results on the demo webpage, which you are welcome to review.

### D.3  MORE DIFFICULT DATASETS WILL BE ATTEMPTED IN THE FUTURE.

The average length of audio-visual speech data is considerably shorter than that of audio-only speech data, potentially making it easier to train. As part of our ongoing efforts, we will develop longer and more complex audio-visual speech translation datasets, aiming to enhance the robustness of Talking Head Translation.

