# OpenReview forum: "TransFace: Unit-Based Audio-Visual Speech Synthesizer for Talking Head Translation"
_ICLR.cc/2024/Conference — ICLR 2024 Conference Desk Rejected Submission_

### Official Review · Reviewer_d3Xc · 2023-10-23

**Soundness:** 3 good
**Presentation:** 3 good
**Contribution:** 3 good
**Rating:** 8
**Confidence:** 3

**Summary:**

This paper presents a new audio-visual speech synthesizer for talking head translation (audio in e.g. English -> audio & video in e.g. Spanish). Instead of relying on a cascade of models (e.g. STT -> translation -> TTS -> Speech to Video) it presents a single model that, without any intermediate mapping to audio or text, produces audio and video in parallel from the original audio stream via a speech-to-unit translation model (S2UT) and unit-to-audio-visual-speech synthesizer (Unit2Lip) which is the first to generate both of these modalities in parallel. The authors also introduce a Bounded Duration Predictor to predict the duration of each unit and ensure that the duration of the generated video is the same as the original video. The authors train AV-resynthesis on LRS2 and A2AV translation on LRS3-T. They find that this model achieves SOTA results and outperforms Cascaded models on MOS, and is competitive on BLEU (but not better - this is understandable since these cascaded models are trained on a lot more text data, as the authors mention), while being around 4x faster due to the parallel AV generation. The authors illustrate some examples of translation towards the end of the paper.

**Strengths:**

Overall, I think this paper has valuable strengths. The proposed method makes sense as an alternative to cascaded models which are obviously suboptimal due to accumulated errors and non-parallelizable generation of audio and video. The architecture seems quite standard but reasonable. The training procedure seems adequate and the introduction of a bounded duration predictor is clever, and helps solve a non-trivial issue in this field. The draft is well-written and clear in most parts and the figures are clear and helpful in understanding what this method is trying to achieve. The results are clear and well-presented, and the authors are upfront about the method's limitations (weaker BLEU compared to cascaded models, for example). The discussion is insightful and welcome, and the conclusions are valid.

**Weaknesses:**

Although the results are sufficient to show that this model is effective, I would appreciate more ablations. These could be on the loss coefficients, for example, or some ablations on the architectural choices (since the architecture as a whole is effectively taken for granted) would be welcome to have a bit more insight into why these components were chosen. The use, for example, of other discrete units as an intermediate representation (the units from EnCodec, or its many variants, or quantized versions of other recent SSL methods) could be explored and would help the readers understand why HuBERT was chosen here specifically. Table 4 should be move to the appendix in my opinion - it's insightful but does not add enough to justify including it in the main draft.

Although I understood it after a few reads, I don't think the Bound Duration Predictor section (4.2) is particularly well-written. There is, for example, a sentence that I believe must have a typo since it does not seem to make sense "(...) we will first the frame represented
by 0.2 in 2.2 will be discarded due to its low weight and the sequence of input discrete units can be represented as (...)". Appendix section A is helpful but could also be written in a more eloquent way, in my opinion, and the pseudocode could be a bit more readable.

Typos:
 - Table 1, for FID, lower is better, so the arrow next to it should be pointing down, not up.
- Page 7 - "Implement details" should be "Implementation details"

**Questions:**

-  Do the authors plan to release code for 1. training 2. inference and 3. pre-trained models? This would be very valuable for the community as a whole.

---

> ### Author Response · Authors · 2023-11-18
> **Response to Reviewer d3Xc**
>
> Thank you for acknowledging our work in terms of writing, architecture design, experimental discussion, analysis, and the significance of the bounded-duration predictor for talking head translation. We appreciate your valuable feedback, and we have thoroughly considered your comments, making important clarifications in response to the raised issues:
>
> ### 1.**Details of bounded-duration-predictor**
>
> We apologize for the typos in the initial paper. Allow us to re-describe them:
>
> > For instance, when U={u1,u2,u3,u4}, D'={2.2,1.8,2.3,2.7}  and T=10,  the weight of each frame can be denoted as U'={1.0\*u1, 1.0\*u1, 0.2\*u1, 1.0\*u2, 0.8\*u2, 1.0\*u3, 1.0\*u3, 0.3\*u3, 1.0\*u4, 1.0\*u4, 0.7\*u4}, where the weight of 0.2\*u1 is only 0.2. And after enough 10 frames have been selected in order of highest to lowest weight([1.0\*u1,1.0\*u1,1.0\*u2,1.0\*u2,1.0\*u3,1.0\*u3,1.0\*u4,1.0\*u4,0.8\*u2,0.7\*u4,0.3\*u3,~~0.2\*u1~~]), 0.2\*u1 is discarded. The sequence of input discrete units can be represented as U′={u1, u1, u2, u2, u3, u3, u3, u4, u4, u4}.
> >
>
> We will add the revised description and pseudocode to the latest version of the paper.
>
> ### 2.**Why discrete units are extracted from mhubert?** (component selection)
>
> We carefully considered the choice of method for extracting discrete units. As a translation task, our objective is to be multilingual, encompassing not only language pairs with English as the target language (such as es-en and fr-en) but also translations involving other languages as the target language (en-es and en-fr, with some results showcased in the demo-page). This broader scope aims to further demonstrate the effectiveness of our model across diverse languages.
>
> Among various discrete unit schemes (encodec, avhubert, hubert, mhubert, etc.), **only mhubert is publicly available and widely used as a multilingual discrete unit extractor.** Hence, we chose mhubert to extract discrete unit representations. As demonstrated in other research [1], if we want the model to encode languages other than English, it must be pre-trained on the speech of these languages.
>
> Here, we also present a comparison of the clustering effect of avhubert and mhubert on different languages. Notably, there is no difference in performance between the two in English, but their effectiveness varies significantly in other languages. Furthermore, in this paper, we showcase that discrete units based on acoustic-only feature can also be effectively utilized for visual speech synthesis.
>
> |  | En |  | Es |  |
> | --- | --- | --- | --- | --- |
> |  | NMI(↑) | Purity(↑) | NMI(↑) | Purity(↑) |
> | AV-Hubert | **43.7** | **65.8** | 12.8 | 9.1 |
> | m-Hubert | 42.6 | 65.1 | **41.7** | **63.2** |
>
> ### 3.**More** **issues:**
>
> - **About Table 4:** Thank you for your valuable suggestions. Initially, our goal was to assist readers in visualizing the quality of the translation. However, we believe your suggestion is highly valuable, and we will thoughtfully select the more crucial elements to be included in the main paper.
> - **About typos:** We appreciate your careful reading of our work, and we will promptly correct these typos.
> - **About Open Source:** We have organized the code and model and are committed to releasing it upon acceptance. We hope that our paper will be successfully accepted.
>
> We believe that these clarifications and enhancements address the concerns raised by the reviewers and further strengthen the credibility and contribution of our work. We appreciate the opportunity to engage in this dialogue and are committed to refining our paper based on your feedback. If you have any other concerns, we look forward to discussing them further with you.
>
> [1]. Le M, Vyas A, Shi B, et al. Voicebox: Text-guided multilingual universal speech generation at scale[J]. arXiv preprint arXiv:2306.15687, 2023.

---

> > ### Comment · Reviewer_d3Xc · 2023-11-20
> >
> > Thank you for the detailed response. Makes sense why you have chosen m-hubert, I had not considered the fact that this is a multi-lingual scenario and that the original Hubert and AV-Hubert are not multi-lingual. If you could include the ablation above and the short discussion around it in the paper, that would be great.
> >
> > I also appreciate your correction and clarification of the bounded-duration-predictor in response to my and other reviewers' comments. And the fact that you have corrected typos, etc. And the fact that you will open-source the code.
> >
> > The general response is also helpful and adds more detailed MOS results - the inclusion of these in the draft will make the paper a lot stronger since what it's missing at the moment is a few more tables with more detailed results.
> >
> > Since the reviewers have thoroughly addressed my concerns, and, from what I have seen, the other reviewers' concerns in a satisfactory way, and the paper, in my opinion, was already strong, I am happy to raise my score.

---

> > > ### Author Response · Authors · 2023-11-20
> > > **Thanks for your response!**
> > >
> > > Thank you for acknowledging the clarifications we provided on various aspects, such as the use of mhubert, the detailed description of the bounded-duration predictor, and the scoring process of MOS, among others. We are delighted that our interactions have contributed to a better understanding of this work, and we believe that your valuable comments have significantly improved the paper. Please feel free to ask additional questions if you have the time. Thank you for raising the score.

---

### Official Review · Reviewer_Qjzu · 2023-10-25

**Soundness:** 3 good
**Presentation:** 3 good
**Contribution:** 3 good
**Rating:** 6
**Confidence:** 4

**Summary:**

This paper presents work towards translating audiovisual speech from one language to audiovisual speech in different languages.  Rather than the traditional pipeline that applies a sequence of models (ASR > MT > TTS), where errors compound, a direct approach is adopted here via units learned using self-supervision.  This results is better audiovisual speech quality in terms of objective and subjective metrics, and results in >4x improvement in generation.

**Strengths:**

+ The paper is tackling a challenging problem.  Speech to speech translation is an important problem, and adding the visual modality significantly increases the complexity of the problem.
+ The authors have considered both objective and subjective assessment, and the results demonstrate the utility of their approach.
+ The paper builds nicely on prior work, and the use of open data helps with reproducibility.

**Weaknesses:**

- The requirement to impose the duration constraint between the source and the target videos seems like a limitation for translation.  I can easily imagine cases where there is a significant mismatch in the length of the source and the target videos.  This seems to be a restriction because of the jarring effects of the background (which still are present in some of the examples) rather than the translation itself.  For example, the short phrase “bucket list” in English translates to “lista de cosas por hacer antes de morir" in Spanish (although there are shortened forms).

**Questions:**

- I am slightly confused about the significance of the difference between extracting units from only acoustic speech vs. from the acoustic component of audiovisual speech.  In both cases there is only acoustic speech being clustered and so would the units not be equivalent?
- In the example for the bounded duration predictor (Section 4.2) — the predicted sequence has a duration of 9 units (2.2 + 1.8 + 2.3 + 2.7) but should span T=10 units.  The paper refers to the first frame at 0.2 being discarded because of the “low weight”.  Why is the first frame at 0.2, and what weight?  From the example, it looks like you just round the predicted durations to the nearest integer and assume that number of repetitions for the duration, but this is not clear from the wording.
- Should Figure 2(b) be referenced in Section 4.3?
- You should state that a five-point Likert scale is used for the MOS ratings.  Also, although you state what is being measured, it would be useful to have the exact wording of the instructions for the viewers in these tests.  How the viewers are instructed to rate the videos can impact what they are actually looking for in the videos.

---

> ### Author Response · Authors · 2023-11-18
> **Response to Reviewer Qjzu (1/2)**
>
> Thank you for recognizing our work in terms of presentation, task significance, experimental adequacy, practicality, and reproducibility. We are grateful for your valuable input and have taken your comments into full consideration and made important clarifications in response to the issues raised:
>
> ### 1.**The isometric feature is crucial for talking head translation**
>
> The isometric feature is pivotal for talking head translation. We acknowledge the concerns raised by the reviewers regarding equal-length translation, and indeed, we extensively deliberated on this matter during our research. **In contrast to speech translation, video translation encounters a relatively fixed limit on video frame length, especially evident when dubbing a translated movie.** In such instances, the voice actor for translated movies must synchronize the translation to match the original video's duration.
>
> The absence of a duration-bounded module in the video results in noticeable frame skipping, leading to a significant loss of realism (refer to the demo page for results without the bounded-duration predictor). This limitation renders the approach unsuitable for professional scenarios like online meetings and movie translating, where the number of generated video frames must align with the original reference video. **The introduction of the bounded-duration predictor becomes imperative in such cases, despite tradeoffs in other factors, as it effectively satisfies the fundamental requirements of video translation**.
>
> Acknowledging the reviewers' observation that this approach may cause excessive speedups and slow reads, we plan to address this concern in our future work. Specifically, we intend to investigate the vocabulary length of the generated content to further enhance the realism and authenticity of the translated videos.
>
> ### 2.**The detailed evaluation process of MOS**
>
> Our comprehensive MOS scoring process for talking head translation tasks involves gathering scores across four dimensions: translation quality, image quality, synchronization, and overall sensation. For the unit-based talking head generation, we streamline the evaluation to two dimensions: image quality and synchronization. Translation quality assesses the consistency of the translated content with the original sentence, while image quality evaluates the presence of artifacts in the generated image. Synchronization measures the coherence of audio and visual speech, and overall sensation indicates the evaluation of the video's authenticity. Each sample is randomly scrambled and presented to 15 participants for scoring. A composite MOS is then calculated by averaging the scores for the corresponding dimensions, with each dimension scored individually on a scale of 1 (lowest) to 5 (highest).
>
> ### 3.**Details of bounded-duration-predictor**
>
> We apologize for the typos in the initial paper. Allow us to re-describe them:
>
> > For instance, when U={u1,u2,u3,u4}, D'={2.2,1.8,2.3,2.7}  and T=10,  the weight of each frame can be denoted as U'={1.0\*u1, 1.0\*u1, 0.2\*u1, 1.0\*u2, 0.8\*u2, 1.0\*u3, 1.0\*u3, 0.3\*u3, 1.0\*u4, 1.0\*u4, 0.7\*u4}, where the weight of 0.2\*u1 is only 0.2. And after enough 10 frames have been selected in order of highest to lowest weight([1.0\*u1,1.0\*u1,1.0\*u2,1.0\*u2,1.0\*u3,1.0\*u3,1.0\*u4,1.0\*u4,0.8\*u2,0.7\*u4,0.3\*u3,~~0.2\*u1~~]), 0.2\*u1 is discarded. The sequence of input discrete units can be represented as U′={u1, u1, u2, u2, u3, u3, u3, u4, u4, u4}.
> >
>
> We will add the revised description and pseudocode to the latest version of the paper.

---

> > ### Author Response · Authors · 2023-11-18
> > **Response to Reviewer Qjzu (2/2)**
> >
> > ### 4.**Why discrete units are extracted from acoustic-only speech:**
> >
> > We carefully considered the choice of method for extracting discrete units. As a translation task, our objective is to be multilingual, encompassing not only language pairs with English as the target language (such as es-en and fr-en) but also translations involving other languages as the target language (en-es and en-fr, with some results showcased in the demo-page). This broader scope aims to further demonstrate the effectiveness of our model across diverse languages.
> >
> > Among various discrete unit schemes (encodec, avhubert, hubert, mhubert, etc.), **only mhubert is publicly available and widely used as a multilingual discrete unit extractor**. Hence, we chose mhubert to extract discrete unit representations. As demonstrated in other research [1], if we want the model to encode languages other than English, it must be pre-trained on the speech of these languages.
> >
> > Here, we also present a comparison of the clustering effect of avhubert and mhubert on different languages. Notably, there is no difference in performance between the two in English, but their effectiveness varies significantly in other languages. Furthermore, in this paper, **we showcase that discrete units based on acoustic-only feature can also be effectively utilized for visual speech synthesis.**
> >
> > |  | En |  | Es |  |
> > | --- | --- | --- | --- | --- |
> > |  | NMI(↑) | Purity(↑) | NMI(↑) | Purity(↑) |
> > | AV-Hubert | **43.7** | **65.8** | 12.8 | 9.1 |
> > | m-Hubert | 42.6 | 65.1 | **41.7** | **63.2** |
> >
> > ### 5.**More** **issues:**
> >
> > - **Add image references:** Thank you for your suggestion, we will add image references in the corresponding positions to enhance readability.
> >
> > We believe that these clarifications and enhancements address the concerns raised by the reviewers and further strengthen the credibility and contribution of our work. We appreciate the opportunity to engage in this dialogue and are committed to refining our paper based on your feedback. If you have any other concerns, we look forward to discussing them further with you.
> >
> > [1]. Le M, Vyas A, Shi B, et al. Voicebox: Text-guided multilingual universal speech generation at scale[J]. arXiv preprint arXiv:2306.15687, 2023.

---

> ### Author Response · Authors · 2023-11-21
> **Isometric translation has become a standard necessity in video translation applications.**
>
> We express our sincere gratitude to the reviewers for their thorough consideration of our work. In order to further underscore the significance of isometric translation in video translation tasks, we conducted an investigation into several websites that have already implemented video translation:
>
> - Video Translation of ***HeyGen***: https://labs.heygen.com/video-translate
> - AI Dubbing & Video Translator of ***ElevenLabs:*** https://elevenlabs.io/dubbing
> - Video Translator of ***Rask AI:*** https://www.rask.ai/tools/video-translator
>
> Remarkably, we found that **all of these platforms have adopted isometric translation as a fundamental requirement for video translation**. As demonstrated by the outcomes of experiments of our own demo page, while variable-duration translation may yield viable results, it unmistakably compromises authenticity. Therefore, **isometric translation is not only advisable but has also been widely recognized as a fundamental prerequisite for video translation applications**.
>
> We trust that this additional survey provided above will aid in clarifying the original intent of isometric translation and contribute to your better understanding and recognition of our work. If you have any further questions or require clarification, please feel free to engage in a discussion with us!

---

> ### Author Response · Authors · 2023-11-22
> **Looking forward to further feedback from Reviewer Qjzu**
>
> Dear Reviewer Qjzu,
>
> Thanks again for your comments. We would like to kindly remind you that we tried our best to address the concerns you raised.
>
> As the end of the author-reviewer discussion period is approaching, we would be grateful if we could hear your feedback regarding our answers to the reviews. We would be happy to discuss in detail if you have additional comments about our paper.
>
> Best regards, Authors

---

> > ### Comment · Reviewer_Qjzu · 2023-11-23
> >
> > Apologies for the delayed reply -- thank you for the detailed response to my questions/concerns.  I have no further questions.

---

> > > ### Author Response · Authors · 2023-11-23
> > > **Thanks for your response!**
> > >
> > > Thank you for acknowledging the clarifications we've provided across different aspects. We are delighted that our interactions have contributed to a deeper understanding of this work, and we believe that your valuable comments have significantly enhanced the quality of this paper.

---

### Official Review · Reviewer_hZGE · 2023-10-31

**Soundness:** 3 good
**Presentation:** 3 good
**Contribution:** 3 good
**Rating:** 6
**Confidence:** 3

**Summary:**

This work proposes TransFace to tackle the task of talking head translation: translates a source video into a target video with underlying audio language properly translated and lip-sync correctly. Traditionally, the translation process is done in a cascade manner where multiple components such as ASR, TTS, Wav2Lip are connected, which requires the pipeline to generate audio first, and then generate video based on audio. This work proposes a simpler pipeline that combines S2U and Unit2Lip where the first component translates the source language into target discrete units, and the second component synthesizes audio and video simultaneously. This approach achieves better synchronization and inference speed.

**Strengths:**

This work proposes using target discrete units to synthesize target audio and target video at the same time, which helps the synchronization and inference speed.
The experiment contains evaluation from many different aspects and the results look convincing.

**Weaknesses:**

Other than adapting the discrete units and duration normalization, many components and approaches seem to be identical to the previous works (i.e. Wav2Lip).

Section 4.3, 4.4 describe the main model and should include more details. for example, it was not clear to me how a and v are computed in Synchronicity Loss. There are also some ambiguous descriptions in a few places, for example, In section 4.2. the author claims that the S2UT model decodes the phoneme sequence. however, If I understand correctly, the proposed approach is predicting discrete units, not phoneme units. why it is decoding phonemes?

**Questions:**

what's the importance of imposing isometric conditions? it seems natural to me that different languages might get different durations depending on the contents. Imposing the same duration condition might lead to unnatural video generation.

during the generation, does the top frame part also get modified to some extent or it is kept the same as the input?

---

> ### Author Response · Authors · 2023-11-18
> **Response to Reviewer hZGE**
>
> Thank you for acknowledging our work in terms of its soundness, presentation, contribution, and experimental adequacy, and for providing valuable feedback. We have thoroughly considered your comments and have made significant clarifications and improvements in response to the raised issues：
>
> ### 1.**The isometric feature is crucial for talking head translation**
>
> The isometric feature is pivotal for talking head translation. We acknowledge the concerns raised by the reviewers regarding equal-length translation, and indeed, we extensively deliberated on this matter during our research. **In contrast to speech translation, video translation encounters a relatively fixed limit on video frame length, especially evident when dubbing a translated movie.** In such instances, the voice actor for translated movies must synchronize the translation to match the original video's duration.
>
> The absence of a duration-bounded module in the video results in noticeable frame skipping, leading to a significant loss of realism (refer to the demo page for results without the bounded-duration predictor). This limitation renders the approach unsuitable for professional scenarios like online meetings and movie translating, where the number of generated video frames must align with the original reference video. **The introduction of the bounded-duration predictor becomes imperative in such cases, despite tradeoffs in other factors, as it effectively satisfies the fundamental requirements of video translation**.
>
> Acknowledging the reviewers' observation that this approach may cause excessive speedups and slow reads, we plan to address this concern in our future work. Specifically, we intend to investigate the vocabulary length of the generated content to further enhance the realism and authenticity of the translated videos.
>
> ### 2.**The introduction of discrete units and bounded duration predictor meets the need for talking head translation.**
>
> Thank you for acknowledging our proposal to integrate discrete units and a bounded duration predictor into talking head translation. **We take pride in being the first to introduce these two designs to the talking head translation task, effectively addressing the challenging issue of the need for equal-length translation in this context.** This innovation enables the parallel synthesis of audio-video-speech, thereby enhancing translation speed and synchronization. Instead of implementing numerous intricate changes to other modules, we have opted to reuse the most basic model of talking head generation. **This decision aims to underscore the modules we propose specifically for the talking head translation task, providing the community with a concise and easy-to-understand framework.**
>
> ### 3.**More** **issues:**
>
> - **Synchronicity Loss:** The synchronization loss requires a pre-trained expert model [1] to compute the synchronization-match (P_sync) between audio speech and video speech. This computed synchronization metric is then used to constrain the synchronization between the two during the training of the talking head generation model.
> - **Top frame part:**
>
>     In the talking head generation task, the majority of methods utilize the upper half of the face as a reference for the current frame. This approach exhibits slight pixel changes during the generation process but generally maintains consistency with the original image.
>
> - **Confusing Expressions: Predicting discrete units, not phoneme units**. We decode discrete units, not phonemes, and we apologize for the misleading expression used here. In the decoding process, there is a specific mapping correlation between the discrete unit and the pronunciation, leading us to use the term "phoneme" in this context. We acknowledge the need to revise this expression in the latest version of the paper.
>
> We believe that these clarifications and enhancements address the concerns raised by the reviewers and further strengthen the credibility and contribution of our work. We appreciate the opportunity to engage in this dialogue and are committed to refining our paper based on your feedback. If you have any other concerns, we look forward to discussing them further with you.
>
> [1]. Chung J S, Zisserman A. Out of time: automated lip sync in the wild[C]//Computer Vision–ACCV 2016 Workshops: ACCV 2016 International Workshops, Taipei, Taiwan, November 20-24, 2016, Revised Selected Papers, Part II 13. Springer International Publishing, 2017: 251-263.

---

> > ### Comment · Reviewer_hZGE · 2023-11-21
> >
> > Thanks authors for the detailed comments! The response helps me understand more about the context and I do not have any further questions.

---

> > > ### Author Response · Authors · 2023-11-21
> > > **Thanks for your response!**
> > >
> > > Thank you for acknowledging the clarifications we've provided across different aspects. We are delighted that our interactions have contributed to a deeper understanding of this work, and we believe that your valuable comments have significantly enhanced the quality of this paper. If you have any further questions or need additional clarification, please feel free to ask. We appreciate your attention to the details.

---

> > > ### Author Response · Authors · 2023-11-21
> > > **An additional survey on Isometric translation for video translation tasks！**
> > >
> > > We sincerely appreciate the thorough review conducted by the reviewers. Given you have also had concerns about isometric translations **before**, we would like to present additional useful research we have conducted on most websites that have achieved video translation (presumably in a cascading manner) functionality:
> > >
> > > - Video Translation by ***HeyGen***: https://labs.heygen.com/video-translate
> > > - AI Dubbing & Video Translator by ***ElevenLabs:*** https://elevenlabs.io/dubbing
> > > - Video Translator by ***Rask AI:*** https://www.rask.ai/tools/video-translator
> > >
> > > Remarkably, we found that **all these platforms have adopted isometric translation as a fundamental requirement for video translation**. As demonstrated by the outcomes of our experiments on our demo page, while variable-duration translation may yield viable results, it unmistakably compromises authenticity. Therefore, isometric translation is not only advisable but has also been widely recognized as a fundamental prerequisite for video translation applications. We believe that the additional survey provided above will further aid in clarifying the original intent of isometric translation and contribute to your better understanding and recognition of our work. If you have any further questions or require clarification, please feel free to engage in a discussion with us!

---

### Official Review · Reviewer_hwnr · 2023-11-01

**Soundness:** 2 fair
**Presentation:** 2 fair
**Contribution:** 2 fair
**Rating:** 5
**Confidence:** 3

**Summary:**

This paper proposes a model for talking head translation, TransFace, which can directly translate audio-visual speech into audio-visual speech in other languages. It
consists of a speech-to-unit translation model to convert audio speech into discrete
units and a unit-based audio-visual speech synthesizer, Unit2Lip, to re-synthesize
synchronized audio-visual speech from discrete units in parallel. The model improves synchronization and boosts inference speed.

**Strengths:**

Talking head translation has many practical applications but has not been systematically studied by prior work. The proposed method, with a duration predictor adapted to isometric setting, achieves better synchronization between audio and video. The proposed model has better inference speed compared to baseline systems.

**Weaknesses:**

1. The proposed method lacks novelty in general. The unit-based approach has been widely used in speech synthesis, both in translation (e.g., [1]) and audio-visual setting (e.g., [2]). Duration prediction is standard in speech synthesis and has also been widely used in unit-based generation (e.g., [1]).
2. The proposed method is evaluated on a constrained benchmark based on synthetic speech with limited video length. It is unclear how it compares to simple cascaded baselines under more challenging scenario. There is inconsistency between the BLEU score and MOS score in the translation results (row 5 vs. 8, Table 2) when compared to the baselines.

[1]. Lee et al., 2021 Textless speech-to-speech translation on real data.

[2]. Hsu et al., 2023 ReVISE: Self-Supervised Speech Resynthesis with Visual Input for Universal and Generalized Speech Enhancement

**Questions:**

Which reference set is used for calculating the FID statistics (the mean & standard deviation), and what model is employed for FID computation?

---

> ### Author Response · Authors · 2023-11-18
> **Response to Reviewer hwnr**
>
> Thank you for acknowledging the Talking Head Translation task in our paper and providing valuable feedback. We have thoroughly considered your comments and implemented important clarifications and improvements to address the raised issues:
>
> ### **1. The First Audio Discrete Units Based Talking Head Generation:**
>
> We appreciate the recognition of our work as the first exploration into talking head translation using audio discrete units:
>
> > *Talking head translation has many practical applications but has not been studied by prior work.*
> >
>
>  As correctly highlighted, prior works you mentioned primarily focused on speech-to-speech translation [1] or only utilized visual speech as input[2]. Our contribution lies in introducing **a novel approach that employs audio discrete units specifically for talking head synthesis.** The parallel generation of highly synchronized audio-visual speech, coupled with the introduction of a bounded duration-predictor for enhanced realism, distinguishes our work from existing research in this domain.
>
> ### **2. Prioritizing Translation Quality Over Voice Cloning:**
>
> It is noteworthy that the speech-to-speech translation works [1] you mentioned did not incorporate voice cloning. **In the realm of translation tasks, I contend that maintaining the consistency of the translated content is more crucial than introducing additional engineering operations for voice cloning** (please also note that the speech part is not our primary contribution).
>
> Additionally, our approach is highly extensible, facilitating seamless integration into various speech-to-speech translation tasks, including [1], among others. Should these efforts subsequently incorporate voice cloning, **our work can be synergistically combined with them to achieve parallel audio-visual speech translation with voice cloning.**
>
> ### **3. Evaluation Metrics:**
>
> - **Authenticity of BLEU Testing Texts:**
>
>     We want to clarify that **the texts used for BLEU testing are not machine-translated**. English, being the only language possessing audio-video speech in our study, is paired with Es-En and Fr-En for experimentation. **The ground-truth text of the translation results are the text from the original LRS3 dataset**, professionally annotated and verified by the official TED, ensuring credibility. Because of the same concern as yours, we purposely did not choose en-es and en-fr for qualitative validation as discussed in Appendix D.1, and only showed the corresponding results in the demo-page.
>
> - **FID Metric Details:**
>
>     To assess the generation quality of the unit-based face synthesizer, we use the FID metric, following the evaluation process of wav2lip[3]. Real video content from LRS2 serves as a reference to gauge the quality of video synthesis on LRS2, employing the InceptionV3 model.
>
> - **Comprehensive Evaluation of Talking Head Translation:**
>
>     We have provided detailed MOS scores for talking head translation tasks, evaluating on four dimensions: translation quality, image quality, synchronization, and overall sensation. **Despite the high translation quality achieved by ASR+NMT+TTS, it is evident that the overall satisfaction is compromised due to jitter caused by repeated references to the reference frames**. With the incorporation of the bounded-duration predictor, TransFace+bounded attains competitive results, notably outperforming other methods in terms of synchronization and overall sensation.
>
>     |  | translation quality | image quality | synchronization | overall sensation | Mean |
>     | --- | --- | --- | --- | --- | --- |
>     | ST+TTS+Wav2Lip | 3.78±0.05 | 4.03±0.08 | 3.66±0.04 | 3.57±0.03 | 3.76±0.05 |
>     | ASR+NMT+TTS+Wav2Lip | **4.23±0.06** | 4.11±0.07 | 4.12±0.08 | 4.18±0.11 | 4.16±0.08 |
>     | Translatotron2+Wav2Lip | 2.79±0.09 | 3.98±0.12 | 3.68±0.07 | 2.91±0.20 | 3.34±0.12 |
>     | S2ST+Wav2Lip | 4.03±0.08 | 4.05±0.04 | 4.02±0.09 | 3.86±0.07 | 3.99±0.07 |
>     | TransFace(ours) | 4.19±0.08 | 4.08±0.07 | **4.28±0.04** | 3.93±0.05 | 4.12±0.06 |
>     | TransFace(ours)+bounded | 4.17±0.06 | **4.16±0.06** | **4.28±0.05** | **4.39±0.11** | **4.25±0.07** |
>
> We believe that these clarifications and enhancements address the concerns raised by the reviewers and further strengthen the credibility and contribution of our work. We appreciate the opportunity to engage in this dialogue and are committed to refining our paper based on your feedback. If you have any other concerns, we look forward to discussing them further with you.
>
> [1]. Lee et al., 2021 Textless speech-to-speech translation on real data.
>
> [2]. Hsu et al., 2023 ReVISE: Self-Supervised Speech Resynthesis with Visual Input for Universal and Generalized Speech Enhancement
>
> [3]. KR Prajwal, Rudrabha Mukhopadhyay, Vinay P Namboodiri, and CV Jawahar. A lip sync expert is
> all you need for speech to lip generation in the wild.

---

> > ### Comment · Reviewer_hwnr · 2023-11-21
> >
> > Thank you for addressing my comments. Could you please provide a few more clarifications?
> > 1. Are evaluation data machine translated or human annotated? My question is about whether **the translation pair** come from an MT system or annotated by human, rather than just **the English text**. The  LRS3-T dataset (Huang et al., 2023),  used in the paper, is constructed by "converting the transcribed English text from LRS3 (Afouras et al., 2018b) into target language using cascaded neural machine translation (NMT) and text-to-speech (TTS) systems" according to [1] (Section 4.1.1). The BLEU score in almost all systems, which includes baselines, are very high (>50 for almost all rows in Table 2), which is potentially related to this fact. This makes it unclear how the proposed method works in real translation setting.
> > 2. On the novelty of duration prediction, Lee et al. 2021 used a very similar approach for predicting the duration of HuBERT units in the target language for speech-to-speech translation. Could you elaborate on how the proposed _bounded-duration-predictor_ differs from that, methodology-wise? Also is there an ablation on keeping the repeating units, which can in principle eliminate the need of  duration model while preserving the length of video (approximately though not 100% guaranteed)?
> > 3. Why are LSE-C and LSE-D not measured in different systems under A-to-AV (Table 2)?

---

> > > ### Author Response · Authors · 2023-11-21
> > > **Further response to Reviewer hwnr (1/n)**
> > >
> > > Hello, thank you very much for taking the time out of your busy schedule to reply to me, and I hope I've resolved your confusion about voice cloning and some Evaluation Metrics details. I will continue to address your questions next:
> > >
> > > # The Synthesized Speech is Widely Used in S2ST and Extensive Comparisons Sufficiently Demonstrate TransFace.
> > >
> > > As you pointed out, the LRS3-T dataset for Language X speech was constructed using a methodology similar to the widely used datasets CVSS-C[1] and CVSS-T [1] in the S2ST domain. This involved combining the NMT and TTS modeling levels to create paired speech, offering an effective solution for the challenging task of collecting paired speech. **Researchers in the S2ST field have leveraged this generative speech pair dataset to conduct experiments in various directions, such as "synthetic to real speech"[4,5] or "real to synthetic speech"[2,3,4], validating their proposed speech-to-speech translation methodologies.** In our work, due to the difficulty in collecting the paired audio-visual speech dataset, we could only validate the effectiveness of our proposed method on the LRS3-T dataset. We present qualitative comparison experiments for "Es-En", "En-Es","Fr-En" and "En-Fr" along with two quantitative comparison experiments for "Es-En" and "Fr-En". Unfortunately, we are unable to provide quantitative comparison experiments for En-Es and En-Fr at this time, as there is no highly reliable AVSR model available for Es and Fr languages.
> > >
> > > We believe that the extensive experimental comparisons in this paper sufficiently demonstrate TransFace on talking head translation:
> > >
> > > 1. Under the same settings, **our proposed TransFace and the cascade model are comparable. TransFace+bound achieves higher subjective scores (MOS)**, providing evidence of TransFace's ability in the talking head translation task.
> > > 2. **Some of our key contributions**, such as a 4.35x speedup and 100% isometric translation, **are applicable in real translation scenarios,** **irrespective of whether the source speech is synthesized or not**.
> > > 3. The S2UT module in this paper aligns with previous S2ST works, demonstrating its applicability to real translation scenarios with real human speech as input.
> > > 4. The translation results of **En-Es and En-Fr on the demo page** simulate real translation scenarios, **using real human speech** as input to showcase the effectiveness of our approach **in real translation scenarios**.
> > >
> > > Finally, we analyze why the translation qualitative metric (BLEU) is so excellent, as detailed in Appendix D.3. There are two main reasons:
> > >
> > > 1. **The inherent simplicity of the LRS3-T dataset itself**, characterized by an average length of 3 seconds, stands in contrast to the CVSS-C[1], Europarl-ST[8], and VoxPopuli[9] datasets with average lengths of 5.7s, 10.6s, and 12s, respectively. The simplicity of the LRS3-T dataset contributes to an improvement in the quantitative metric for speech translation (BLEU). In general, the shorter the sentence the easier it is to understand and the higher the BLEU score is likely to be.
> > > 2. Due to the scarcity of open-source audio-visual speech datasets, we are using AV-Hubert, a representative work in the field of audio-visual speech recognition, to perform AVSR on the translated results, but **AV-Hubert is trained on LRS3, resulting in a strong speech recognition performance on the domain of the LRS3 and LRS3-T dataset.** We show here a WER comparison between the AVSR model we used and the ASR model used in other S2ST work, where the WER results for the ASR model are from [7].
> > >
> > > |  | Dataset | En (WER %) | Es (WER %) | Fr (WER %) |
> > > | --- | --- | --- | --- | --- |
> > > | ASR model | VoxPopuli |  14.2 | 15.5 | 18.5 |
> > > | AV-Hubert | LRS3-T | 1.3 | - | - |
> > >
> > > [1] Jia Y et al. CVSS corpus and massively multilingual speech-to-speech translation[J]. LREC 2022.
> > >
> > > [2] Li X, Jia Y, Chiu C C. Textless direct speech-to-speech translation with discrete speech representation ICASSP 2023-2023 IEEE International Conference on Acoustics, Speech and Signal Processing  ICASSP 2023.
> > >
> > > [3] Shi J, Tang Y, Lee A, et al. Enhancing Speech-To-Speech Translation with Multiple TTS Targets ICASSP 2023.
> > >
> > > [4] Huang R, Liu J, et al. Transpeech: Speech-to-speech translation with bilateral perturbation[J]. ICLR 2022.
> > >
> > > [5] Dong Q, Huang Z, Xu C, et al. PolyVoice: Language Models for Speech to Speech Translation[J]. arXiv preprint arXiv 2023.
> > >
> > > [6] Shi B, Hsu W N, Lakhotia K, et al. Learning audio-visual speech representation by masked multimodal cluster prediction[J]. ICLR 2022.
> > >
> > > [7] Lee A, Gong H, Duquenne P A, et al. Textless speech-to-speech translation on real data. ACL 2021.
> > >
> > > [8] Iranzo-Sánchez J, Silvestre-Cerda J A, Jorge J, et al. Europarl-st: A multilingual corpus for speech translation of parliamentary debates ICASSP 2020.
> > >
> > > [9] Wang C, Riviere M, Lee A, et al. VoxPopuli: A large-scale multilingual speech corpus for representation learning, semi-supervised learning and interpretation. ACL 2021.

---

> > > ### Author Response · Authors · 2023-11-22
> > > **Looking forward to further feedback from Reviewer hwnr**
> > >
> > > Dear Reviewer hwnr,
> > >
> > > Thanks again for your comments. We would like to kindly remind you that we tried our best to address the concerns you raised.
> > >
> > > As the end of the author-reviewer discussion period is approaching, we would be grateful if we could hear your feedback regarding our answers to the reviews. We would be happy to discuss in detail if you have additional comments about our paper.
> > >
> > > Best regards, Authors

---

> ### Author Response · Authors · 2023-11-20
> **Voice cloning was not performed in most of the previous S2ST work.**
>
> Hello reviewer hwnr, **we've noticed that there may be some inherent impressions regarding the S2ST task, particularly with a focus on voice cloning**. To ensure a fair assessment and prevent any potential bias, we've curated demo pages from past speech-to-speech translation works for your consideration:
>
> 1. ****Textless Speech-to-Speech Translation on Real Data [1]:**** https://facebookresearch.github.io/speech_translation/textless_s2st_real_data/index.html
> 2. ****Direct Speech-to-Speech Translation With Discrete Units [2]:**** https://facebookresearch.github.io/speech_translation/direct_s2st_units/index.html
> 3. **TranSpeech: Speech-to-Speech Translation With Bilateral Perturbation [3]:** https://transpeech.github.io/
> 4. **SeamlessM4T-Massively Multilingual Multimodal Machine Translation [4]:** https://seamless.metademolab.com/
>
> These demos highlight the emphasis on **improving the consistency of the translated speech content, prioritizing this aspect over the realization of voice cloning**. We trust that these demonstrations will provide valuable insights for your evaluation. If you have any further questions or require clarification, please feel free to engage in a discussion with us!
>
> [1] Lee A, Gong H, Duquenne P A, et al. Textless speech-to-speech translation on real data[J]. arXiv preprint arXiv:2112.08352, 2021.
>
> [2] Lee A, Chen P J, Wang C, et al. Direct speech-to-speech translation with discrete units[J]. arXiv preprint arXiv:2107.05604, 2021.
>
> [3] Huang R, Liu J, Liu H, et al. Transpeech: Speech-to-speech translation with bilateral perturbation[J]. arXiv preprint arXiv:2205.12523, 2022.
>
> [4] Barrault L, Chung Y A, Meglioli M C, et al. SeamlessM4T-Massively Multilingual & Multimodal Machine Translation[J]. arXiv preprint arXiv:2308.11596, 2023.

---

> ### Author Response · Authors · 2023-11-21
> **Further response to Reviewer hwnr (2/n), n=2**
>
> # **Difference between bounded-duration-predictor and naive duration-predictor.**
>
> 1. The naive duration predictor directly rounds up the predicted duration sequence after predicting the duration length for each discrete unit. This approach often leads to a mismatch between the length of the synthesized translation result and the original audio.
> 2. Making adjustments to the naive duration predictor by normalizing it and then rounding it (as in step 1 of the bounded-duration-predictor) allows for some degree of duration control. However, this method still encounters issues when dealing with more or fewer frames. For instance, in scenarios like:
>     - Case 1: *D*′=[1.566,1.666,1.766],*T*=5 (Predicted frames = 6, one extra frame)
>     - Case 2: *D*′=[1.55,1.6,1.55,1.65,1.65],*T*=8 (Predicted frames = 10, two extra frames)
>     - Case 3: *D*′=[2.2,1.8,2.3,2.7],*T*=10 (Predicted frames = 9, one frame less).
> 3. Our proposed bounded-duration-predictor module addresses these challenges by **implementing an efficient duration allocation method as detailed in Appendix A.** This method ensures effective control of the total duration of the discrete unit sequence, resulting in 100% isometric translation.
>
> # **An Ablation on Keeping the Repeating Units**
>
> I'm not sure I fully understand what you're saying, but I'm guessing that you might want an ablation experiment of TranFace without bounded-duration-predictor in Table 3. In fact, its length metrics are all essentially the same as the S2ST+Wav2Lip approach, as both use the same naive duration-predictor, which does not allow for overall modulation of duration:
>
> |  | LR | LC@5↑ | LC@10↑ | LC@20↑ |
> | --- | --- | --- | --- | --- |
> | S2ST+Wav2Lip | +0.055(1.055) | 15.80 | 31.87 | 57.46 |
> | TransFace | +0.053(1.053) | 15.79 | 31.83 | 57.48 |
> | TransFace+Early Stop | -0.051(0.949) | 77.63 | 83.79 | 93.76 |
> | TransFace+bounded | **0.000(1.000)** | **100.00** | **100.00** | **100.00** |
>
> If my understanding deviates from your question in any way, feel free to continue asking, and I'm here to address any further doubts you may have.
>
> # The synchronization of A-to-AV results is consistent with the unit-based talking head generation model.
>
> Throughout the process of talking head translation (A-to-AV), the synchronization of audio-visual speech in the generated results is dependent on the specific audio-visual speech synthesis model used. Put simply, **the synchronization metric can be cross-referenced with the outcomes from different talking head generation models detailed in Table 1**. To address your curiosity, we promptly conducted an assessment of the synchronization in all the generated audio-visual speech and present the comparison in the following table:
>
> |  | LSE-C↑ | LSE-D↓ |
> | --- | --- | --- |
> | ST+TTS+Wav2Lip | 6.310 | 7.325 |
> | ASR+NMT+TTS+Wav2Lip | 6.296 | 7.298 |
> | Translatotron2+Wav2Lip | 6.284 | 7.341 |
> | S2ST+Wav2Lip | 6.312 | 7.270 |
> | TransFace(ours)+bounded | **7.291** | 7.113 |
> | TransFace(ours) | 7.276 | **7.084** |
>
> ## Summary
>
> Thank you for your prompt response and your interest in this paper. Your insightful questions have contributed to a more comprehensive understanding of the details associated with this work. I trust that my responses have addressed all of your inquiries. If you have any further questions, please do not hesitate to ask.

---

> ### Comment · Reviewer_hwnr · 2023-11-22
>
> Thank you for addressing my comments.
> 1. On real data vs. synthesized data, the real concern is about the extremely high BLEU score: 97.04 in using ground-truth video. For such high BLEU score, the mapping between source speech and target text is almost deterministic, which isn't the case for real translation (e.g., the BLEU of Whisper for Es-En is typically between 20 and 40). As is already pointed out either in the rebuttal or the paper (Table 2), there exists a trade-off between translation quality and synchronicity for cascaded models (ASR+NMT+TTS+Wav2Lip) vs. the proposed approach (Transface). As the measurement of translation quality is directly impacted by the evaluation data, It is a bit unclear to me how such trade-off looks like when using a more challenging dataset. In addition, there does exist real multilingual data for Es-En and Fr-En such as Multilingual TEDx [1], which has already been used in audio-visual setting in several prior works [2] (recognition), [3] (translation). Is there a reason of not using that? As to the contribution on speed, I did acknowledge that in the review.
> 2. On duration prediction, it's a natural extension to the duration prediction of Lee et al. 2021 when the length of target speech is given. This isometric assumption makes sense in using the LRS3-T data. However, in general, the length of target speech can vary much from the length of source speech. Under a setting where its length is unknown, how does one predict the target length and how does the prediction error impacts the performance? Or simply how much value does the isometric assumption add compared to a simple cascaded system (e.g., ASR+NMT+TTS+Wav2Lip) without such constraint? Again this is related to the data issue mentioned above.
> 3. The question on synchronicity is resolved, thank you.
>
> [1]. Salesky, E. et al. The Multilingual TEDx Corpus for Speech Recognition and Translation. In Proceedings of the 22nd Annual Conference of International Speech Communication Association.
>
> [2]. Pingchuan M et al., Visual Speech Recognition for Multiple Languages in the Wild. Nature Machine Intelligence
>
> [3]. Mohamed A et al., MuAViC: A Multilingual Audio-Visual Corpus for Robust Speech Recognition
> and Robust Speech-to-Text Translation.

---

> > ### Author Response · Authors · 2023-11-22
> > **Looking forward to prompt feedback from Reviewer hwnr.**
> >
> > We have endeavored to provide quick and comprehensive explanations to your inquiries, and we trust that our responses, coupled with real examples of video translation platforms, will dispel any doubts and enhance your understanding of the motivation behind this work.
> >
> > As the end of the author-reviewer discussion period is approaching, if you have any additional questions or require further clarification, please feel free to ask. We would greatly appreciate a prompt response.
> >
> > Best regards, Authors

---

> > > ### Comment · Reviewer_hwnr · 2023-11-23
> > >
> > > Thank you for addressing my comments. Overall my concerns on the experiment (FID, synchronicity in translation) have been resolved, thus I will increase the score. However, I do think there is an issue on model benchmarking. Using more realistic audio-visual data, especially for evaluation, will make several comparisons (e.g., cascaded system vs. TransFace, isometric vs. non-isometric) more convincing.

---

> > > > ### Author Response · Authors · 2023-11-23
> > > > **Thanks for your response!**
> > > >
> > > > Thank you once again for acknowledging the clarifications we've provided! Your suggestions have significantly bolstered the coherence and comprehensiveness of this work.
> > > >
> > > > Furthermore, the incorporation of four translation directions in our experiments ("es-en," "fr-en," "en-es," and "en-fr") effectively **underscores the significance of our approach in video translation tasks**. The experimental setup for En-Es and En-Fr even **consistently aligns with real talking head translation scenarios**.
> > > >
> > > > Nevertheless, we also **agree with your valuable insight** that “*Using more realistic audio-visual data, especially for evaluation, will make several comparisons more convincing.*” Although **we are unable to conduct relevant experiments due to the limitations of the current datasets**, we eagerly anticipate the availability of more pertinent paired audio-visual speech translation datasets in the future!

---

> ### Author Response · Authors · 2023-11-22
> **Further response to Reviewer hwnr (1/n)**
>
> Thank you for your response. Allow me to provide some clarifications in response to your inquiries:
>
> Firstly, I want to underscore that the experiment with ID=0 in Table 2 (id=0, BLEU=97.04) involved calculating BLEU directly from **the audio-visual speech recognition results of the target English audio-visual speech** in LRS3-T. Our intention was to demonstrate the additional errors that may be introduced by utilizing the audio-visual speech recognition model. It's crucial to note that this process represents a mapping from target language speech to target language text, rather than which you described *"mapping between source speech and target text."*
>
> Additionally, there are Three primary reasons for choosing LRS3-T over Multilingual TEDx:
>
> 1. The average length of Multilingual TEDx is 6.5 seconds, while all current audio-visual speech datasets are approximately 3 seconds. **Performing translation on this heavily excessive length of speech could result in significant reference frame reuse or audio acceleration.** This not only diminishes the authenticity of the translation result for the speaking head but also introduces interference that can adversely impact the subjective evaluation of the translation outcome.
> 2. LRS3-T has real target audio-visual speech, it can provide **better quantitative comparison**.
> 3. LRS3-T allows for **the simulation of the real talking head translation scenarios** (the results of En-Es and En-Fr) with higher persuasive power.
>
> Furthermore, it's worth noting that you mention muavic, which is another multilingual audio/video speech dataset. We have indeed examined muavic but found its performance to be less impressive. Even when evaluated on the dataset used for training, its Audio-Visual Speech Recognition (AVSR) WER are only 16.2% (Es) and 19.0% (Fr). In contrast, the ASR and AVSR models utilized for validation in speech-to-speech translation are required to perform well on their respective training datasets. For instance, the ASR model in [1] achieved a Word Error Rate (WER) of 2.0% on LJSpeech, and the AVSR model we employed achieved a WER of 1.5% on LRS3. Consequently, we believe that utilizing **the AVSR model of language X from muavic could potentially introduce a significant amount of error** into the results presented in this paper.
>
> [1] Lee A, Gong H, Duquenne P A, et al. Textless speech-to-speech translation on real data. ACL 2021.

---

> > ### Author Response · Authors · 2023-11-22
> > **Further response to Reviewer hwnr (2/n)**
> >
> > ## **The length of the translated video is constrained based on the source video**
> >
> > Before addressing your second question, allow me to answer this one:
> >
> > > Under a setting where its length is unknown, how does one predict the target length?
> > >
> >
> > I would like to reiterate our setup: **Given a source language audio speech, our goal is to translate it into an audio-visual speech in the target language.** To simulate video translation and avoid potential video discontinuity in the case of variable-length translation, it is crucial that **the length of the generated audio-visual speech matches that of the source video (the reference video)**. You can learn about video translation task settings from the video translation platforms we surveyed (https://openreview.net/forum?id=71oyMJiUm2&noteId=uV5tsVEcTc).
> >
> > **In real application scenarios, the length of the translated video can be directly constrained based on the length of the source audio-visual speech, ensuring isometric translation.** Specifically, in this work, where there is no audio-visual speech available in our X language, we use the corresponding English visual speech as the reference video for synthesis. Although its length may not precisely match that of the original X-language speech, **it is sufficient to demonstrate the effectiveness of the 100% equal-length translation that we have achieved.**

---

> ### Author Response · Authors · 2023-11-22
> **Further response to Reviewer hwnr (3/n), n=3**
>
> Thank you for recognizing our proposed bounded-duration-predictor, who brings more novel features to duration-predictor. I think your current problem can be summarized as the meaning of isometric translation, which has been discussed in discussions with other reviewers (https://openreview.net/forum?id=71oyMJiUm2&noteId=FZva2rAT19) and in the general response (https://openreview.net/forum?id=71oyMJiUm2&noteId=rNyqlgFpnm).
>
> **In contrast to speech translation, video translation encounters a relatively fixed limit on video frame length, especially evident when dubbing a translated movie.** In such instances, the voice actor for translated movies must synchronize the translation to match the original video's duration.
>
> The absence of a duration-bounded module in the video results in noticeable frame skipping, leading to a significant loss of realism (refer to the demo page for results without the bounded-duration predictor). This limitation renders the approach unsuitable for professional scenarios like online meetings and movie translating, where the number of generated video frames must align with the original reference video. **The introduction of the bounded-duration predictor becomes imperative in such cases, despite tradeoffs in other factors, as it effectively satisfies the fundamental requirements of video translation**.
>
> We also conducted a survey of existing video translation platforms (https://openreview.net/forum?id=71oyMJiUm2&noteId=uV5tsVEcTc). Remarkably, we found that **$\textcolor{red}{\text{all these platforms have adopted isometric translation as a fundamental requirement for video translation}}$**. As demonstrated by the outcomes of our experiments on our demo page, while variable-duration translation may yield viable results, it unmistakably compromises authenticity. Therefore, isometric translation is not only advisable but has also been widely recognized as a fundamental prerequisite for video translation applications.
>
> Acknowledging the reviewers' observation that this approach may cause excessive speedups and slow reads, we plan to address this concern in our future work. Specifically, we intend to investigate the vocabulary length of the generated content to further enhance the realism and authenticity of the translated videos. We have incorporated a discussion on the limitations of isometric translations in Appendix D.1 to underscore the significance of isometric translations while also acknowledging the potential associated limitations.

---

### Author Response · Authors · 2023-11-18
**Response to all reviewers （1/2）**

We express our gratitude to the reviewers for recognizing various aspects of our work:

- **the significance of the talking head translation** [hwnr,hZGE,Qjzu,d3Xc]
- **the validity of the methodology** [hwnr,hZGE,Qjzu,d3Xc]
- **the clarity and accessibility of the writing** [hZGE, Qjzu,d3Xc]
- **the adequacy of the experiments** [hZGE, Qjzu,d3Xc]
- **the soundness of the analysis** [Qjzu,d3Xc]
- **the ease of reproducibility** [Qjzu]
- **the importance of isometric translation for talking head translation** [d3Xc].

We are also thankful for the valuable suggestions provided by the reviewers, and in the following, we will highlight and clarify a few essential points:

### 1.**The isometric feature is crucial for talking head translation**

The isometric feature is pivotal for talking head translation. We acknowledge the concerns raised by the reviewers regarding equal-length translation, and indeed, we extensively deliberated on this matter during our research. **In contrast to speech translation, video translation encounters a relatively fixed limit on video frame length, especially evident when dubbing a translated movie.** In such instances, the voice actor for translated movies must synchronize the translation to match the original video's duration.

The absence of a duration-bounded module in the video results in noticeable frame skipping, leading to a significant loss of realism (refer to the demo page for results without the bounded-duration predictor). This limitation renders the approach unsuitable for professional scenarios like online meetings and movie translating, where the number of generated video frames must align with the original reference video. **The introduction of the bounded-duration predictor becomes imperative in such cases, despite tradeoffs in other factors, as it effectively satisfies the fundamental requirements of video translation**.

Acknowledging the reviewers' observation that this approach may cause excessive speedups and slow reads, we plan to address this concern in our future work. Specifically, we intend to investigate the vocabulary length of the generated content to further enhance the realism and authenticity of the translated videos.

### 2.**The detailed evaluation process of MOS**

Our comprehensive MOS scoring process for talking head translation tasks involves gathering scores across four dimensions: translation quality, image quality, synchronization, and overall sensation. For the unit-based talking head generation, we streamline the evaluation to two dimensions: image quality and synchronization. Translation quality assesses the consistency of the translated content with the original sentence, while image quality evaluates the presence of artifacts in the generated image. Synchronization measures the coherence of audio and visual speech, and overall sensation indicates the evaluation of the video's authenticity. Each sample is randomly scrambled and presented to 15 participants for scoring. A composite MOS is then calculated by averaging the scores for the corresponding dimensions, with each dimension scored individually on a scale of 1 (lowest) to 5 (highest). Here, we present the MOS (Mean Opinion Score) results for both talking head translation and unit-based talking head generation, and we will incorporate this into the latest version:

- MOS for Talking Head Translation:


    |  | translation quality |  image quality |  synchronization | overall sensation | Mean |
    | --- | --- | --- | --- | --- | --- |
    | ST+TTS+Wav2Lip | 3.78±0.05 | 4.03±0.08 | 3.66±0.04 | 3.57±0.03 | 3.76±0.05 |
    | ASR+NMT+TTS+Wav2Lip | **4.23±0.06** | 4.11±0.07 | 4.12±0.08 | 4.18±0.11 | 4.16±0.08 |
    | Translatotron2+Wav2Lip | 2.79±0.09 | 3.98±0.12 | 3.68±0.07 | 2.91±0.20 | 3.34±0.12 |
    | S2ST+Wav2Lip | 4.03±0.08 | 4.05±0.04 | 4.02±0.09 | 3.86±0.07 | 3.99±0.07 |
    | TransFace(ours) | 4.19±0.08 | 4.08±0.07 | **4.28±0.04** | 3.93±0.05 | 4.12±0.06 |
    | TransFace(ours)+bounded | 4.17±0.06 | **4.16±0.06** | **4.28±0.05** | **4.39±0.11** | **4.25±0.07** |
- MOS for Unit-based Talking Head Generation:

    |  |  image quality |  synchronization | Mean |
    | --- | --- | --- | --- |
    | U2S+LipGAN | 2.89±0.25 | 2.39±0.21 | 2.64±0.23 |
    | U2S+Wav2Lip | **4.01±0.20** | 3.93±0.24 | 3.92±0.22 |
    | Unit2Lip(ours) | 3.95±0.24 | **4.01±0.24** | **3.98±0.24** |
    | Audio(GT)+Wav2lip | 4.18±0.32 | 4.12±0.18 | 4.15±0.25 |
    | Video(GT) | **4.33±0.13** | **4.13±0.11** | **4.23±0.12** |

---

> ### Author Response · Authors · 2023-11-18
> **Response to all reviewers （2/2）**
>
> ### 3.**Details of bounded-duration-predictor**
>
> We apologize for the typos in the initial paper. Allow us to re-describe them:
>
> > For instance, when U={u1,u2,u3,u4}, D'={2.2,1.8,2.3,2.7}  and T=10,  the weight of each frame can be denoted as U'={1.0\*u1, 1.0\*u1, 0.2\*u1, 1.0\*u2, 0.8\*u2, 1.0\*u3, 1.0\*u3, 0.3\*u3, 1.0\*u4, 1.0\*u4, 0.7\*u4}, where the weight of 0.2\*u1 is only 0.2. And after enough 10 frames have been selected in order of highest to lowest weight([1.0\*u1,1.0\*u1,1.0\*u2,1.0\*u2,1.0\*u3,1.0\*u3,1.0\*u4,1.0\*u4,0.8\*u2,0.7\*u4,0.3\*u3,~~0.2\*u1~~]), 0.2\*u1 is discarded. The sequence of input discrete units can be represented as U′={u1, u1, u2, u2, u3, u3, u3, u4, u4, u4}.
> >
>
> In the specific implementation, we meticulously devised algorithms to attain the bounded duration goal:
>
> 1. After normalization, the duration ***D*** can be computed as the allocated duration ***D’*** corresponding to each token based on the target length ***T***.
>
>     > ***D′*** = Normalize(***D***)×T
>     >
> 2. Following the rounding method, it is converted to an integer predicted duration ***PRED***.
>
>     > ***PRED*** = Clamp(Round(***D′***),min = 1)
>     >
> 3. Calculate the difference ***DIFF***  between the predicted duration ***PRED*** and the allocated duration ***D’*** for each token.
>
>     > ***DIFF*** = ***D′***−***PRED***
>     >
> 4. Determine whether the predicted duration ***PRED*** still needs adjustment in the number of frames. If an increase is required, select the highest-weighted difference ***DIFF*** from the sequence for its duration+1. Conversely, if a decrease is needed, select the lowest-weighted difference ***DIFF*** for its duration-1.
>
>     > ***ADD*** =Zeroes()
>     >
>     >if Sum(***PRED***) > ***T*** then
>     >
>     >    ***INDEX*** =TopK(***−DIFF***,k =Sum(***PRED***)−***T***)
>     >
>     >    ***ADD*** [***INDEX*** ]= −1 ;
>     >
>     >else
>     >
>     >    ***INDEX*** =TopK(***DIFF***,k =***T*** −Sum(***PRED***))
>     >
>     >    ***ADD*** [***INDEX*** ]= 1
>     >
>     >end
>     >
>     >***OUT*** =***PRED*** +***ADD***
>     >
>
> Let's revisit the previous example for illustration, when ***U***={u1,u2,u3,u4}, ***D'***={2.2,1.8,2.3,2.7}  and ***T***=10:
>
> >**Step 1:** ***D'***=[2.2,1.8,2.3,2.7], ***T***=10
> >
> >**Step 2:** ***PRED***=[2,2,2,3]
> >
> >**Step 3:** ***DIFF***=[0.2,-0.2,0.3,-0.3]
> >
> >**Step 4:** Since SUM(***PRED***)=9<10, for the largest ***T***-SUM(***PRED***)=1 corresponding token in ***DIFF***, its duration+1, resulting in ***OUT***=[2,2,2+1,3] = [2,2,3,3]
>
> We will add the revised description and pseudocode to the latest version of the paper.
>
> We express our sincere gratitude to all the reviewers for their diligence and invaluable suggestions. We look forward to any further insights or inquiries that may arise from our work.

---

### Author Response · Authors · 2023-11-22
**General Response to All Reviewers**

We would like to thank the reviewers for their constructive reviews! Following the comments and suggestions of reviews, we have revised the manuscript, and the revised parts are marked in blue. Here we summarize the revision as follows:

- **[d3Xc, Qjzu]:** We have elaborated on the implementation process of the Bounded-Duration-Predictor in detail in Section 4.2 (Summary and Appendix A). Additionally, we have included more examples to enhance the reader's comprehension.
- **[hwnr, Qjzu]**: We have augmented the paper's experimental details by introducing the detailed MOS scoring process and a comparison of MOS scores for each dimension in Appendix B.3.
- **[Qjzu, d3Xc]**: We have added a discussion in Appendix C.3 explaining the rationale behind using mhubert for extracting discrete units.
- **[Qjzu, hZGE]:** We have incorporated a discussion on the limitations of isometric translations in Appendix D.1 to underscore the significance of isometric translations while also acknowledging the potential associated limitations.
- **[hwnr, hZGE]**: We have included a comprehensive description and references to some technical details (FID and Synchronicity Loss, etc.).
- **[Qjzu]:** We have included the relevant image references in Subsection 4.3.
- **[d3Xc]**: We have relocated the translation results of Fr-En from the main text to Appendix C.4.

Thanks again for the reviewers' great efforts and valuable comments, which have improved the soundness of the manuscript. We have carefully addressed the main concerns and provided detailed responses to each reviewer. We hope you will find the responses satisfactory. We would be grateful if we could hear your feedback regarding our answers to the reviews.